Atom Search Optimization: a comprehensive review of its variants, applications, and future directions

El-Shorbagy Mohammed A. 1
Bouaouda Anas 2 anas.bouaouda-etu@etu.univh2c.ma
Abualigah Laith 3 4
Hashim Fatma A. 5 6
1 Department of Mathematics, College of Science and Humanities in Al-Kharj, Prince Sattam bin Abdulaziz University , Al-Kharj , Saudi Arabia
2 Faculty of Science and Technology, Hassan II University of Casablanca , Mohammedia , Morocco
3 School of Engineering and Technology, Sunway University Malaysia , Petaling Jaya , Malaysia
4 Centre for Research Impact & Outcome, Chitkara University Institute of Engineering and Technology, Chitkara University , Punjab , India
5 Faculty of Engineering, Helwan University, Egypt , Cairo , Egypt
6 Applied Science Research Center, Applied Science Private University , Amman , Jordan
Angiulli Giovanni
Electronic publication date: 2025 Feb 28
Publication date: 2025
Volume: 11
Electronic Location ID: e2722
Received 2024 Oct 3; Accepted 2025 Jan 30
Copyright: © 2025 El-Shorbagy et al.
Copyright year: 2025
Copyright holder: El-Shorbagy et al.
License: This is an open access article distributed under the terms of the Creative Commons Attribution License, which permits unrestricted use, distribution, reproduction and adaptation in any medium and for any purpose provided that it is properly attributed. For attribution, the original author(s), title, publication source (PeerJ Computer Science) and either DOI or URL of the article must be cited.
License URL: https://creativecommons.org/licenses/by/4.0/

Keywords: Metaheuristics, Atom search optimization, Engineering problems, Applications, Hybridization, Global optimization

Funding: Prince Sattam bin Abdulaziz University 2024/RV/06 This work was supported by Prince Sattam bin Abdulaziz University through the project number (2024/RV/06). The funders had no role in study design, data collection and analysis, decision to publish, or preparation of the manuscript.

==============================
The Atom Search Optimization (ASO) algorithm is a recent advancement in metaheuristic optimization inspired by principles of molecular dynamics. It mathematically models and simulates the natural behavior of atoms, with interactions governed by forces derived from the Lennard-Jones potential and constraint forces based on bond-length potentials. Since its inception in 2019, it has been successfully applied to various challenges across diverse fields in technology and science. Despite its notable achievements and the rapidly growing body of literature on ASO in the metaheuristic optimization domain, a comprehensive study evaluating the success of its various implementations is still lacking. To address this gap, this article provides a thorough review of half a decade of advancements in ASO research, synthesizing a wide range of studies to highlight key ASO variants, their foundational principles, and significant achievements. It examines diverse applications, including single- and multi-objective optimization problems, and introduces a well-structured taxonomy to guide future exploration in ASO-related research. The reviewed literature reveals that several variants of the ASO algorithm, including modifications, hybridizations, and multi-objective implementations, have been developed to tackle complex optimization problems. Moreover, ASO has been effectively applied across various domains, such as engineering, healthcare and medical applications, Internet of Things and communication, clustering and data mining, environmental modeling, and security, with engineering emerging as the most prevalent application area. By addressing the common challenges researchers face in selecting appropriate algorithms for real-world problems, this study provides valuable insights into the practical applications of ASO and offers guidance for designing ASO variants tailored to specific optimization problems.

Introduction

Optimization problems are widespread in real-world engineering domains, encompassing challenges such as path, structural, and parameter optimization, all characterized by inherent complexity (Velasco, Guerrero & Hospitaler, 2024). Over the past few decades, various optimization methods have been developed to address these challenges, including steepest descent, dynamic programming, and linear programming. However, many traditional optimization techniques rely heavily on gradient information, making them ineffective for solving non-differentiable function optimization problems (Toaza & Esztergár-Kiss, 2023). Moreover, conventional methods often struggle with multi-modal problems, frequently converging to local optima due to numerous local minima. Addressing such complex optimization problems remains a key focus in intelligent optimization. In response to these challenges, metaheuristic algorithms have emerged as effective alternatives to traditional methods, demonstrating superior performance and finding widespread application across diverse optimization domains. Their ability to navigate complex search spaces and overcome the limitations of conventional techniques has established them as invaluable tools in modern optimization practices (Mandour, Gamal & Sleem, 2024).

In recent decades, metaheuristic methods have proven remarkably effective in addressing engineering optimization challenges. These methods employ random search principles to explore the search space for optimal solutions (Sarhani, Voß & Jovanovic, 2023). Metaheuristics are favored over gradient-based techniques for their ability to perform global exploration, independence from derivatives, adaptability, and simplicity. Metaheuristic algorithms typically follow a similar methodology to find optimal solutions (Tzanetos & Blondin, 2023). Initially, a vector of random solutions is generated within the feasible space. New solutions are then produced through specialized operators. These existing and newly generated solutions are evaluated and selected through various mechanisms. This iterative process continues until the optimal solution is identified (Tzanetos & Blondin, 2023).

In broad terms, the process of a metaheuristic method can be divided into two main phases. The first phase is exploration, which involves thoroughly searching the feasible domain to identify potential regions where the optimal solution might reside. This phase extensively explores the solution space (Morales-Castañeda et al., 2020). Following the exploration phase, the algorithm transitions into the exploitation phase. This phase focuses on delving deeper into the promising regions identified during exploration, aiming to pinpoint the optimal solution within these areas (Morales-Castañeda et al., 2020). These two phases inherently compete during optimization. Therefore, an effective method must strike a delicate balance between exploration and exploitation. This balance allows the algorithm to identify the global optimal solution while avoiding entrapment in local optima (Jia & Lu, 2024).

Metaheuristic methods can be broadly categorized into two primary classes: non-nature-inspired and nature-inspired methods (Bouaouda & Sayouti, 2022). While non-nature-inspired methods have introduced new tools across various domains, they often exhibit limitations in practical applications. In contrast, nature-inspired methods have emerged as dominant players in optimization designs across diverse fields as research progresses. These methods offer several advantages, including more robust search capabilities, fewer parameter requirements, and lower demands on problem characteristics (Abdel-Basset, Abdel-Fatah & Sangaiah, 2018). In recent years, nature-inspired methods have found widespread application in a multitude of domains, including renewable energy (Saber & Salem, 2023), image processing (Hashim et al., 2024), feature selection (Pirozmand et al., 2023), mobile cloud computing (Saemi et al., 2023), medical diagnostics (Hosseinabadi et al., 2022), and various other complex environments.

Nature-inspired methods are categorized into various classes based on the sources of their inspiration (see Fig. 1), including evolutionary-based, human-based, physics-based, mathematics-based, chemistry-based, and swarm-based algorithms (Jia et al., 2023b). Evolutionary-based techniques involve transforming the primary problem into a population of individuals and improving solutions by simulating mutation, crossover, and selection. Notable examples include the Artificial Immune System (AIS) (Greensmith, Whitbrook & Aickelin, 2010), Biogeography-Based Optimization (BBO) (Simon, 2008), Black Widow Optimization (BWO) (Hayyolalam & Kazem, 2020), Bull Optimization Algorithm (BOA) (Oğuz, 2015), Stochastic Fractal Search (SFS) (Salimi, 2015), Differential Evolution (DE) (Storn & Price, 1997), Genetic Programming (GP) (Koza, 1994), Genetic Algorithm (GA) (Holland, 1992), Backtracking Search Algorithm (BSA) (Civicioglu, 2013), Evolutionary Strategy (ES) (Yao, Liu & Lin, 1999), and Human Felicity Algorithm (HFA) (Kazemi & Veysari, 2022), among others.

Figure 1 Metaheuristic algorithm categories.

The second category consists of human-based algorithms primarily inspired by human behavior. An illustrative example in this category is Teaching-Learning-Based Optimization (TLBO) (Rao, Savsani & Vakharia, 2011), which leverages a pedagogical framework inspired by teacher-student interactions to find optimal solutions in various optimization problems effectively. Other well-known techniques in this category include the Mountaineering Team-Based Optimization (MTBO) (Faridmehr et al., 2023), Gold Rush Optimizer (GRO) (Zolf, 2023), Chief Executive Officer Election Algorithm (CEOA) (Jia et al., 2023a), City Council Evolution (CCE) (Pira, 2023), Ali Baba and the Forty Thieves (AFT) (Braik, Ryalat & Al-Zoubi, 2022), Sewing Training-Based Optimization (STBO) (Dehghani, Trojovská & Zuščák, 2022), Growth Optimizer (GO) (Zhang et al., 2023b), Political Optimizer (PO) (Askari, Younas & Saeed, 2020), Driving Training-Based Optimization (DTBO) (Rehman et al., 2023), Great Wall Construction Algorithm (GWCA) (Guan et al., 2023), Exchange Market Algorithm (EMA) (Ghorbani & Babaei, 2014), among others.

The third category includes physics-based approaches, which involve simulations based on physical laws. An example is the Gravitational Search Algorithm (GSA) (Rashedi, Nezamabadi-Pour & Saryazdi, 2009), which is founded on the concepts of mass interaction and gravitational forces. The Simulated Annealing (SA) (Van Laarhoven et al., 1987) algorithm draws inspiration from metallurgical annealing processes, where controlled heating and cooling of material enhance crystal size and reduce defect density. Other notable algorithms in this category include Young’s Double-Slit Experiment Optimizer (YDSE) (Abdel-Basset et al., 2023a), Snow Ablation Optimizer (SAO) (Deng & Liu, 2023), Kepler Optimization Algorithm (KOA) (Abdel-Basset et al., 2023c), Light Spectrum Optimizer (LSO) (Abdel-Basset et al., 2022), Special Relativity Search (SRS) (Goodarzimehr et al., 2022), Plasma Generation Optimization (PGO) (Kaveh, Akbari & Hosseini, 2020), Momentum Search Algorithm (MSA) (Dehghani & Samet, 2020), Multi-Verse Optimizer (MVO) (Mirjalili, Mirjalili & Hatamlou, 2016), Atomic Search Optimization (ASO) (Zhao, Wang & Zhang, 2019a), PID-based Search Algorithm (PSA) (Gao, 2023), and others.

Mathematics-based methods constitute the fourth class of metaheuristic algorithms, leveraging features derived from mathematical theories, formulas, and functions to enhance the computational efficiency of optimization techniques. Among these methods, the Arithmetic Optimization Algorithm (AOA) (Abualigah et al., 2021) stands out as a renowned optimizer, drawing inspiration from the inherent distribution characteristics of basic arithmetic operations, including Addition (A), Subtraction (S), Division (D), and Multiplication (M). Other notable examples in this category include the Sinh Cosh Optimizer (SCHO) (Bai et al., 2023), Quadratic Interpolation Optimization (QIO) (Zhao et al., 2023), Exponential Distribution Optimizer (EDO) (Abdel-Basset et al., 2023b), weIghted meaN oF vectOrs (INFO) (Ahmadianfar et al., 2022), RUNge kutta optimizer (RUN) (Ahmadianfar et al., 2021), Sine Cosine Algorithm (SCA) (Mirjalili, 2016), Geometric Mean Optimizer (GMO) (Rezaei et al., 2023), Hyperbolic Sine Optimizer (HSO) (Thapliyal & Kumar, 2024), Gradient-based optimizer (GBO) (Ahmadianfar, Bozorg-Haddad & Chu, 2020), and others.

Chemistry-based methods constitute another category of optimization algorithms, often drawing inspiration from the principles underlying chemical reactions, such as Brownian motion, molecular radiation, and molecular reactions. Among the algorithms in this group, some of the most commonly utilized ones include Thermal Exchange Optimization (TEO) (Kaveh & Dadras, 2017), Ions Motion Optimization (IMO) (Javidy, Hatamlou & Mirjalili, 2015), Gases Brownian Motion Optimization (GBMO) (Abdechiri, Meybodi & Bahrami, 2013), Ray Optimization (RO) (Kaveh & Khayatazad, 2012), Smell Agent Optimization (SAO) (Salawudeen et al., 2021), Artificial Chemical Reaction Optimization Algorithm (ACROA) (Alatas, 2011), Chemical Reaction Optimization (CRO) (Lam & Li, 2012), Atomic Orbital Search (AOS) (Azizi, 2021), and others.

The final category encompasses swarm-based algorithms inspired by the collective behaviors observed in biological populations, particularly the communicative and cooperative behaviors found in plant and animal populations. Swarm-based methods exhibit diversity and breadth in their approaches. Among these, Particle Swarm Optimization (PSO) (Kennedy & Eberhart, 1995) stands out as one of the most renowned and earliest extended swarm-based methods, mimicking the swarming behavior of birds. Several other notable swarm-based methods include Red Panda Optimization (RPO) (Hadi, Mohammad & Štěpán, 2023), Genghis Khan Shark (GKS) (Hu et al., 2023), Grey Wolf Optimizer (GWO) (Mirjalili, Mirjalili & Lewis, 2014), Golden Jackal Optimization (GJO) (Chopra & Ansari, 2022), Nutcracker Optimization Algorithm (NOA) (Abdel-Basset et al., 2023d), Meerkat Optimization Algorithm (MOA) (Xian & Feng, 2023), Bald Eagle Search (BES) (El-Shorbagy et al., 2024), Pied Kingfisher Optimizer (PKO) (Bouaouda et al., 2024), and others.

Despite the number and variety of metaheuristic algorithm classes, the No Free Lunch Theorem (NFL) (Wolpert & Macready, 1997) asserts that no single algorithm can effectively solve all optimization problems. An algorithm that performs well on a benchmark test set may not necessarily be suitable for specific real-world engineering challenges, and vice versa. Advances in science and technology continually introduce more complex optimization problems, often revealing the limitations of existing metaheuristic algorithms. To address these challenges, improving current algorithms is essential to enhance their performance for specific problem types (Hashim et al., 2024). Researchers have also developed novel strategies to overcome the common limitations of metaheuristic algorithms, broadening their applicability (Jia & Lu, 2024). Furthermore, designing entirely new metaheuristic algorithms represents a promising avenue for tackling increasingly complex optimization tasks.

In 2019, Zhao, Wang & Zhang (2019a) introduced the Atom Search Optimization (ASO) algorithm, a physics-inspired metaheuristic algorithm designed to solve global optimization problems. Drawing on the principles of molecular dynamics, ASO simulates the behavior of atoms, modeling their movements and interactions within a solution space (Zhao, Wang & Zhang, 2019a). Each atom represents a potential solution, with its mass serving as an indicator of the solution’s quality. By leveraging attractive and repulsive forces, ASO enables atoms to explore and exploit the search space effectively, guiding the algorithm toward optimal solutions (Zhao, Wang & Zhang, 2019a). ASO’s innovative approach offers several notable advantages. Its simplicity and flexibility make it adaptable to various optimization tasks. Furthermore, the algorithm demonstrates computational efficiency, enabling rapid convergence to high-quality solutions. Notably, ASO’s stochastic nature eliminates the need for derivative information, simplifying its implementation and broadening its applicability. These combined strengths position ASO as an emerging tool for solving complex optimization problems across diverse domains.

The simplicity and effectiveness of ASO have attracted considerable research interest, leading to numerous studies focused on enhancing its operational capabilities. Beyond theoretical advancements, ASO and its variants have effectively addressed real-world challenges across various domains, including environmental optimization, complex system modeling, clustering, data mining, healthcare, and engineering. This versatility highlights ASO’s capability to handle complex and evolving problems. The continuous evolution of ASO, driven by incremental enhancements and innovative hybrid methods, underscores active research and indicates a promising future for ASO in theoretical and practical applications.

This article comprehensively reviews the ASO algorithm, covering its foundational principles, variants, and implementations for addressing constrained, unconstrained, single-objective, multi-objective, and large-scale global optimization problems, as well as real-world challenges. This review aims to deepen the understanding of ASO’s functionality and performance by examining its algorithmic structure and search behavior. We explore various parameter adaptation strategies, topologies, and the effectiveness of ASO variants across different optimization scenarios, including its hybridization with other methods. Our analysis draws on literature from 2019 to 2024, sourced from Scopus and Web of Science databases, emphasizing simplicity and clarity in ASO research, focusing on high-impact studies. Additionally, this review highlights ASO’s robustness and the modifications proposed to overcome its limitations while also listing significant contributions from previous studies and featuring key works published in leading publishers such as Springer, Elsevier, IEEE, and others.

The contributions of this review article are outlined below. A comprehensive explanation of ASO, including its mathematical foundations, provides readers with a solid understanding of how the algorithm tackles optimization challenges;

In-depth analysis of ASO variants, focusing on models tailored for multi-objective optimization, modifications, and hybrid approaches that demonstrate the algorithm’s versatility in addressing complex optimization problems;

A comprehensive overview of the various application areas where the ASO has been successfully employed to tackle complex and nonlinear problems in scientific and engineering design;

Suggestions for future research directions, potential areas for improvement in the ASO algorithm, and recommendations for new application domains.

The remainder of this review article is organized as shown in Fig. 2. “Review Methodology” outlines the methodology used for the review. “The Popularity and Growth of ASO in the Literature” analyzes ASO’s growth in the literature, including the number of ASO-related articles, citations, authors, countries, institutions, and research topics since 2019. “Overview of Atom Search Optimization” presents background information on the conceptual features and structure of the classical ASO algorithm. “Recent Variants of Atom Search Optimization” offers an in-depth review of various ASO adaptations, while “Applications of the Atom Search Algorithm” discusses ASO applications across different fields. “Open Source Software of ASO” introduces open-source ASO software to support researchers. “Critical Analysis and Discussion” provides a critical analysis of ASO’s performance, identifying research gaps and limitations. Finally, “Conclusion and Further Works” concludes the article by summarizing the study and its key findings while offering future research directions in the ASO field.

Figure 2 Review article organization.

Review methodology

This section outlines the methodology employed to conduct a comprehensive review of the ASO algorithm. The approach includes procedures for collecting, analyzing, interpreting, and synthesizing information to achieve an in-depth understanding of the subject. A keyword-based search was conducted, targeting articles that mentioned either “Atom Search Optimization” OR “ASO” in their titles or abstracts to ensure relevance. The search was limited to publications from 2019 to September 2024, focusing on recent advancements. Using the Scopus (https://www.scopus.com) and Web of Science (https://www.webofscience.com) databases, 410 relevant articles were identified. This extensive dataset provided a solid foundation for the subsequent analysis and evaluation of the ASO algorithm.

After removing duplicates and filtering out irrelevant articles, we identified 223 articles directly related to Atom Search Optimization. Of these, 73 articles were excluded because they mentioned ASO only as a comparison algorithm without providing a detailed analysis. Articles consisting of editorial comments, reviews, or letters, as well as those not written in English, were also excluded. This rigorous selection process ensured that only high-quality and relevant research was included in the final analysis. Figure 3 illustrates the comprehensive collection and selection process, emphasizing the meticulous approach to ensure the review’s accuracy and reliability.

Figure 3 Process of collecting articles.

A total of 150 articles that met the inclusion criteria were collected and analyzed from two perspectives. First, the articles were categorized based on their ASO variants, including modified versions (binary, opposition-based learning, chaotic-based, Lévy-based, adaptive, fractional calculus-based, quantum-based, Laplacian operator-based, crossover-based, and multi-strategy), hybridized approaches, and multi-objective versions. Second, they were reviewed by application areas to identify trends across various fields, such as engineering, healthcare and medical applications, the Internet of Things and communication, clustering and data mining, environmental modeling, and security.

The popularity and growth of aso in the literature

Due to its simplicity and adaptability, the ASO algorithm has seen a significant increase in research and application in recent years. This growth is evident in the rising number of published articles and citations related to ASO. To illustrate this trend, we conducted a comprehensive literature review using the Scopus database. Scopus offers various data retrieval options, including support for regular expressions and advanced query design, which facilitate searches by titles, articles, authors, and temporal criteria. The Scopus search was conducted based on the following criteria: The search query used is “Atom Search Optimization”.

The publication date range is restricted to articles between 2019 and 2024.

The results were collected on 10/09/2024.

Figure 4 illustrates the growth in citations for ASO publications, showing a substantial increase from 9 in 2019 to 144 in 2023, reflecting the algorithm’s rising recognition within the research community. As of September 2024, the citation count had already reached 79, suggesting it may surpass the 2023 total by the end of the year. This exponential rise in citations underscores ASO’s effectiveness and efficiency in solving optimization problems and reinforces its growing impact across various fields.

Figure 4 The number of ASO citations per year.

Figure 5 illustrates the annual growth in ASO publications, reflecting the increasing recognition of ASO as a key algorithm across various fields. Starting with just 10 publications in 2019, the number of ASO-related articles surged to 71 by the end of 2023. As of September 2024, there have already been 23 publications, indicating that the total for the year may match or exceed the 2023 output. This growing trend suggests that ASO is gaining popularity among researchers and is likely to continue.

Figure 5 The number of ASO publications per year.

Figure 6 illustrates the primary mediums for publishing ASO research, with journals emerging as the predominant platform, followed by conferences. This distribution underscores the scholarly community’s preference for journal articles as the main method for advancing knowledge on the ASO algorithm.

Figure 6 The number of ASO publications per type.

Table 1 provides details on the journals contributing to ASO research. “IEEE Access” leads with six publications, followed by “Multimedia Tools and Applications” with five articles. The third position is shared by six journals, each with four publications: “Heliyon”, “International Transactions on Electrical Energy Systems”, “Neural Computing and Applications”, “Journal of Ambient Intelligence and Humanized Computing”, “Sustainability”, and “Wireless Personal Communications”. The complete ranking is presented in the table below.

Table 1 The top 20 journals ranked by number of publications related to ASO.

No.	Journal	Number of articles	
1	IEEE access	6	
2	Multimedia tools and applications	5	
3	Heliyon	4	
4	International transactions on electrical energy systems	4	
5	Neural computing and application	4	
6	Journal of ambient intelligence and humanized computing	4	
7	Sustainability	4	
8	Wireless personal communications	4	
9	Applied soft computing	3	
10	Arabian journal for science and engineering	3	
11	E prime advances in electrical engineering electronics and energy	3	
12	Electrica	3	
13	Energies	3	
14	Expert systems with applications	3	
15	IETE journal of research	3	
16	Knowledge-based systems	2	
17	Energy conversion and management	2	
18	Engineering applications of artificial intelligence	2	
19	Journal of cleaner production	2	
20	Mathematical problems in engineering	2	

Figure 7 illustrates ASO publications by affiliation. Researchers from “Batman University” have published over 17 ASO-related research articles, while “Aswan University” and the “Faculty of Engineering” have each contributed more than seven publications. Other notable institutions include “Prince Sattam Bin Abdulaziz University”, “Zagazig University”, and “Future University in Egypt”, each making significant contributions. This highlights ASO’s growing prominence across various academic institutions and its potential as a powerful method for solving research problems.

Figure 7 The number of ASO publications per affiliation.

In terms of leading authors in ASO research, this review identifies “Ekinci, S”. as the most prominent contributor, with 15 publications where they served as the first author. Following closely are “Izci, D.” with 13 publications and “Goud, B.S.”, “Shareef, H.I.”, “Wang, L.”, “Zhang, Y.”, and “Zhao, W.”, each contributing four publications as the first author. This distribution of contributions is illustrated in Fig. 8, highlighting the key contributors to the ASO algorithm.

Figure 8 The number of ASO publications per author.

Finally, Fig. 9 illustrates the countries where the ASO algorithm has been employed in research. China leads with 158 publications, followed by India and Egypt with 113 and 66 publications, respectively, reflecting their use of ASO to tackle various optimization challenges. Researchers from several other countries have also utilized ASO to advance their research.

Figure 9 The number of ASO publications per country.

Overview of atom search optimization

The following subsections discuss the ASO algorithm’s inspiration, mathematical model, framework, computational complexity, parameters, and limitations.

Inspiration and principle

The theoretical foundation of ASO is inspired by molecular dynamics, a field that studies the structure of matter and atomic interactions at the microscopic scale. ASO employs a search mechanism based on the interaction and constraint forces between pairs of atoms (Zhao, Wang & Zhang, 2019a). Interaction forces, encompassing both repulsive and attractive components, mimic the forces that bind atoms together. Constraint forces, on the other hand, are used to maintain fixed distances between atoms connected by covalent bonds. Throughout the iterative process, atoms interact via these forces. When the distance between two atoms is less than the equilibrium distance, the force is attractive; conversely, it becomes repulsive when it exceeds the equilibrium distance. Additionally, a constraint force acts between each atom and the atom with the highest fitness (Zhao, Wang & Zhang, 2019a).

Mathematical model

Three key mathematical components are defined in the atomic system: interaction force, geometric constraint force, and atomic motion. The details of these components are provided below.

Interaction force

Each atom interacts with every other atom. The overall interaction force applied to the i-th atom in the d-th dimension at iteration t is described by the following equation (Zhao, Wang & Zhang, 2019a):

(1) Fid(t)=∑j∈KbestrandjFijd(t)

where Kbest is a subset of the total atom population, comprising the top K best atoms, and randj is a random number between [0,1]. To balance exploration and exploitation, the value of K gradually decreases over iterations, as described by the equation below (Zhao, Wang & Zhang, 2019a).

(2) K(t)=N−(N−2)×tT

where T is the maximum number of iterations and N represents the total atom population. Additionally, Fid(t) denotes the interaction force, reflecting either negative repulsion or positive attraction forces, and is given by the following expression (Zhao, Wang & Zhang, 2019a).

(3) Fij(t)=−η(t)[2(hij(t))−13−(hij(t))−7]r→ijrij

where η(t) denotes the depth function controlling the intensity of repulsion or attraction at the t-th iteration and is updated as follows (Zhao, Wang & Zhang, 2019a):

(4) η(t)=α(1−t−1T)3e−20tT

where T denotes the maximum number of iterations and α represents the depth weight. The scaled distance between two atoms is defined as follows (Zhao, Wang & Zhang, 2019a):

(5) hij(t)={hmin,rij(t)σ(t) < hminrij(t)σ(t),hmin≤rij(t)σ(t)≤hmaxhmax,rij(t)σ(t) > hmax

where hmax and hmin denote the upper and lower bounds of the scaled distance h, respectively, and are defined as follows (Zhao, Wang & Zhang, 2019a):

(6) h={hmin=g0+g(t)hmax=u

where u is the upper bound value of 1.24 and g0 is the lower bound value of 1.1. g(t) represents the drift factor, which adjusts the algorithm’s focus from exploration to exploitation and is defined as follows (Zhao, Wang & Zhang, 2019a):

(7) g(t)=0.1×sin⁡(π2×tT).

Furthermore, σ(t) represents the collision diameter and is defined as follows (Zhao, Wang & Zhang, 2019a):

(8) σ(t)=‖xij(t),∑j∈Kbestxjd(t)K(t)‖2

where ‖⋅,‖2 denotes the 2-norm, which can be computed using the Euclidean distance.

Geometric constraint force

A geometric constraint is applied to maintain the structure of a polyatomic molecule. This constraint is modeled by considering that each atom is covalently bonded to the most optimal atom. The mathematical expression for the constraint force acting on the i-th atom is given by Zhao, Wang & Zhang (2019a):

(9) Gi(k)=−λ(k)∇θi(k)=−2λ(k)(xi(k)−xbestp(k))

where λ is the Lagrangian multiplier expressed as below (Zhao, Wang & Zhang, 2019a).

(10) λ(k)=βe−20kkmax

where β is the multiplier weigh. Furthermore, θi represents the geometric constraint for the i-th atom mathematically presented as an equation below (Zhao, Wang & Zhang, 2019a).

(11) θi(k)=[|xi(k)−xbestp(k)|2−bi2].

Atomic motion

The atomic motion follows the principles of classical mechanics. The acceleration of atom i in the d-th dimension is determined by the combined effects of interaction forces and geometric constraint forces, as given by the following equation (Zhao, Wang & Zhang, 2019a):

(12) aid(t)=Fid(t)mid(t)+Gid(t)mid(t)

=−α(1−t−1T)3e−20tT∑j∈Kbestrandj[2×(hij(t)13−hij(t)7)mi(t)](xjd(t)−xid(t))‖xi(t),xj(t)‖2

+ βe−20tTxbestd(t)−xid(t)mi(t)

where mid(t) represents the mass of the i−th atom in the d−th dimension at iteration t, and is computed using the following expression (Zhao, Wang & Zhang, 2019a):

(13) mi(t)=Mi(t)∑j=1NMj(t).

In which Mi(t) is calculated as follow (Zhao, Wang & Zhang, 2019a).

(14) Mi(t)=e−Fiti(t)−Fitbest(t)Fitworst(t)−Fitbest(t)

where Fitworst(t) and Fitbest(t) are the fitness values of the worst and best atoms at iteration t, respectively. However, Fiti(t) is the fitness of the i-th atom at iteration t, and. The velocity and position update formulas for the i-th atom in the j-th dimension at iteration t can be mathematically expressed below (Zhao, Wang & Zhang, 2019a).

(15) vid(t+1)=randidvid(t)+aid(t)

(16) xid(t+1)=xid(t)+vid(t+1)

where vi and xi represent the velocity and position of the i-th atom, ai denotes the acceleration, and randi is a random number within [0,1].

All calculations and updates are performed iteratively until the stopping criterion is met. Ultimately, the position and objective function value of the best atom represent the optimal approximation for the problem.

Framework of ASO algorithm

The ASO optimization process begins by generating an initial random set of solutions. During each iteration, atoms adjust their positions and velocities, with the position of the best-performing atom also being updated. The acceleration of each atom is determined by two components: the interaction force from the Lennard-Jones potential (which combines attractive and repulsive forces) and the constraint force from the bond-length potential (which represents a weighted difference between the atom’s position and the position of the best atom). These updates and calculations are repeated iteratively until the termination criteria are satisfied. Lastly, the position and fitness value of the best atom approximates the global optimum. The pseudo-code for the ASO algorithm is outlined in Algorithm 1.

Algorithm 1 Pseudo-code of ASO algorithm.

1: Input: upper and lower bounds of variables (xmax and xmin), dimension of the problem (D), maximum number of iterations (T), number of population (N), β, and α.	
2: Initialize the velocity and position of atoms randomly.	
3: Calculate the fitness of atoms.	
4: Record the best solution.	
5: Initialize iteration count: t=1.	
6: while (t≤T) do	
7:   for i = 1 to N do	
8:     Compute mass by Eqs. (13) and (14).	
9:     Define subset K atoms by Eq. (2).	
10:     Calculate the interaction force by Eq. (1).	
11:     Compute the constraint force by Eq. (9).	
12:     Calculate the acceleration by Eq. (12).	
13:     Update the velocity of the atoms by Eq. (15).	
14:     Update the position of atoms by Eq. (16).	
15:     Check if the new position is within boundaries.	
16:     Evaluate the current fitness.	
17:   end for	
18:    t=t+1.	
19: end while	
20: Output: Return the best solution.	

Complexity analysis of ASO algorithm

The computational complexity of the ASO algorithm depends on the problem dimensionality (D), atom population size (N), and maximum iteration count (T). The primary computational costs arise from the initialization phase, fitness evaluations, memory management, position updates, and acceleration updates. Consequently, the overall computational complexity of the ASO algorithm can be expressed as follows:

(17) O(ASO)=O(Problem definition)+O(Initialization)+O(Functione valuation)+ O(Memory saving)+O(Position updating)+O(Acceleration updating)O(ASO)=O(1)+O(ND)+O(TCN)+O(TN)+O(TND)+O(TND)O(ASO)=O(1+ND+TCN+TN+TND+TND)O(ASO)≅O(TND+TCN).

Parameter sensitivity

The ASO algorithm is notably simple to implement and requires minimal parameter tuning. Aside from the standard parameters such as maximum iterations, atom population size, and problem dimension, only the upper and lower bounds need to be defined. For the function F′, setting g0=1.1, and u=1.24 generally produces good results. The depth and multiplier weights, represented by α and β, respectively, can be empirically adjusted within 0 to 100 for α and 0 to 1 for β (Zhao, Wang & Zhang, 2019a). However, extensive testing suggests optimal performance is typically achieved within the narrower ranges of 40≤α≤60 and 0.1≤β≤0.3 (Zhao, Wang & Zhang, 2019a). These ranges serve as a general guideline, as specific problems may require fine-tuning these parameters. Nevertheless, setting α=50 and β=0.2 offers a reasonable starting point for many applications. Researchers can further adjust these parameter values based on the unique characteristics of their optimization problem to enhance performance.

ASO against state-of-the-art algorithms

This subsection provides a comprehensive assessment of the ASO algorithm, comparing it with other state-of-the-art algorithms, as summarized in Table 2. As shown in the table, ASO exhibits distinct characteristics compared to other algorithms regarding the number of parameters and complexity. However, it also shares specific strengths and weaknesses with these algorithms. The analysis highlights that the basic ASO requires enhancements to improve its capabilities, address its limitations, and achieve superior performance compared to other algorithms.

Table 2 Overall comparison between ASO and state-of-the-art algorithms.

Ref.	Algorithm	Year	Category	Parameter	Complexity	Merits	Limitations	
Zhao, Wang & Zhang (2019b)	Atom search optimization (ASO)	2019	Physics-based	2	O(TND+TCN)	Easy implementation and high Capability	Lower population diversity and may struggle with high-dimensional problems	
Kennedy & Eberhart (1995)	Particle swarm algorithm (PSO)	1995	Swarm-based	4	O(N+N×T)	Simple to implement, fast convergence	Prone to premature convergence, sensitive to parameter tuning	
Mirjalili & Lewis (2016)	Whale optimization algorithm (WOA)	2016	Swarm-based	5	O(TND+TN)	Simple implementation, good exploration	Probability of falling into local optima	
Wang et al. (2022a)	Artificial rabbit optimizer (ARO)	2022	Swarm-based	2	O(TND+TN+N)	Flexibility, stability, good balance between exploration and exploitation	Slow convergence	
Faramarzi et al. (2020)	Marine predators algorithm (MPA)	2020	Swarm-based	3	O(TND+TN)	Effective in handling multimodal problems	Limited in global search capability	
Heidari et al. (2019)	Harris Hawks optimization (HHO)	2019	Swarm-based	2	O(TND+TN+N)	Strong exploitation ability	Convergence may slow in high dimensions	
Gandomi, Yang & Alavi (2013)	Cuckoo search (CS)	2009	Swarm-based	1	O(N+2NT)	Performs well in complex problems	Suffer from slow convergence	
Mirjalili, Mirjalili & Lewis (2014)	Grey wolf optimizer (GWO)	2014	Swarm-based	1	O(N+N×T)	Simple structure, easy to implement	Limited exploration capabilities	
Holland (1992)	Genetic algorithm (GA)	1992	Evolution-based	3	O(TN+N)	Handles discrete and continuous optimization	Evaluation is relatively expensive	
Storn & Price (1997)	Differential evolution (DE)	1997	Evolution-based	2	O(TN+N)	Effective for continuous optimization	May require many function evaluations	
Geem, Kim & Loganathan (2001)	Harmony search (HS)	2001	Human-based	3	O(N+N×T)	Suitable for discrete and continuous problems	May require parameter tuning	
Van Laarhoven et al. (1987)	Simulated annealing (SA)	1987	Physics-based	2	O(1+T)	Good global optimization ability	Slow convergence in large search spaces	
Mirjalili (2016)	Sine cosine algorithm (SCA)	2016	Mathematics-based	1	O(TND+TN+N)	Simple, few parameters, balances exploration and exploitation	May get stuck in local optima	

Limitation of ASO algorithm

ASO demonstrates robust optimization capabilities and has been applied to various engineering challenges. However, it has notable limitations. The velocity update equation in ASO relies exclusively on individual atom velocities and accelerations, which can reduce population diversity and limit information exchange by neglecting the influence of atom positions (Sun et al., 2021a; Wang et al., 2022b). Furthermore, the attractive force in ASO can cause atoms to cluster prematurely, leading to early convergence. Conversely, the repulsive force may be insufficient to help atoms escape local optima in later stages (Joshi et al., 2024). This premature convergence limits exploration, while the inability to escape local optima affects exploitation, ultimately limiting ASO’s effectiveness in solving complex optimization problems. Various variants have been proposed to address these challenges and enhance ASO’s performance, as detailed in the following section.

Recent variants of atom search optimization

The original ASO algorithm was designed for numerical optimization problems, and its effectiveness was assessed using standard benchmark functions, similar to other metaheuristics. Several variants have since been developed to extend ASO’s applicability to a broader range of problems with varying characteristics or structures, including modified, hybrid, and multi-objective versions, as shown in Fig. 10. This section provides a concise overview of each variant, referencing relevant examples from the literature.

Figure 10 Classification of ASO variants.

Modified versions of atom search optimization

Like many algorithms, ASO can face challenges in achieving global optima, such as slow convergence, local optima traps, and an imbalance between exploration and exploitation. To overcome these limitations, researchers have introduced several modifications to the ASO search process. Additionally, ASO has been adapted to handle different search spaces, extending its applicability beyond its original design for continuous domains. This subsection reviews the modified ASO variants, categorizing them based on the types of modifications, with each category discussed in its respective subsection.

Binary ASO

Too & Abdullah (2020a) introduced Binary ASO (BASO), a variant of the ASO algorithm designed for wrapper feature selection tasks. BASO adapts the continuous ASO algorithm to a binary format using eight transfer functions from V-shaped and S-shaped families. Applied to 22 benchmark datasets from the UCI machine learning repository, BASO effectively identifies significant features that improve classification efficiency. Compared to PSO and Binary Differential Evolution (BDE), BASO demonstrated superior classification accuracy with fewer features, highlighting its potential in feature selection applications.

Wu, Ren & Hao (2021) proposed an advanced method for optimizing polyester fiber production by integrating a binary encoding ASO with a feature selection Extreme Learning Machine (ELM) model. The approach uses ASO for feature selection and optimization of the ELM model’s weights, thresholds, and activation functions. By combining binary encoding with distance measures from K-nearest neighbor, the model achieves high accuracy in predicting intrinsic viscosity, a key factor for fiber quality. This method offers substantial improvements in process optimization and is well-suited for industrial applications.

Hammadi & Qasim (2022) introduced SD-BASO, which combines statistical dependence (SD) with Binary ASO (BASO) for enhanced feature selection. This method aims to identify key dataset features, improving classification accuracy in scientific and medical applications. The SD-BASO algorithm utilizes a novel fitness function to select significant features efficiently, showing superior accuracy and efficiency compared to traditional methods, as demonstrated by experiments on multiple datasets.

El Tokhy (2023) proposed a sophisticated method for distinguishing BGO and LSO scintillator crystals in positron emission tomography (PET) to reduce parallax errors. Their approach utilizes a Matlab Simulink model to simulate scintillation pulses from a 22Na source, employing two main algorithms: one for feature extraction using 1D-Walsh ordered Fast Walsh-Hadamard Transform (1WFWHT) and Fast Chebyshev Transform (FCHT), optimized with Binary ASO (BASO), and classification using Adaptive Neuro-Fuzzy Inference System (ANFIS) and naive Bayes classifiers; and another leveraging convolutional neural networks (CNN) for feature extraction and Deep Neural Networks (DNN) for classification. This dual approach significantly enhances accuracy in PET imaging by improving both feature extraction and classification.

Asna, Shareef & Prasanthi (2023a) developed a framework to optimize fast charging station locations and capacities for electric vehicles, addressing challenges like travel times, power losses, and station utilization. The authors used a Binary ASO (BASO) algorithm, demonstrating improved network efficiency and reduced waiting times for EV users.

In other work, Asna, Shareef & Prasanthi (2023b) proposed QBASO, a quantum-inspired binary variant of the ASO algorithm for optimizing EV charging station locations and capacities. QBASO integrates quantum binarization techniques, including quantum gates and bits, to enhance binary search space optimization. Tested on benchmark functions and EV charging station placement, QBASO proved to be an efficient and effective tool for multi-objective infrastructure planning in EV networks.

Vempati, Sharma & Tripathy (2024) developed a method for emotion recognition from EEG signals using multivariate fast iterative filtering (MvFIF) and binary ASO (BASO). Their approach decomposes EEG signals into intrinsic mode functions, extracts various entropy features, and uses BASO for feature space optimization. Classification with a light gradient boosting machine (LGBM) achieved the highest accuracy of 99.50% and 98.79% in cross-subject emotion recognition, demonstrating the effectiveness of combining MvFIF and BASO for accurate emotion detection.

Chaotic ASO

Hekimoğlu (2019) introduced the chaotic ASO (ChASO) algorithm for optimizing FOPID controller parameters in DC motor speed control systems. By incorporating chaotic sequences, ChASO enhances ASO’s ability to avoid local minima and improve convergence rates. The effectiveness of ChASO was demonstrated on benchmark test suites and compared with other controllers like GWO-FOPID and IWO-PID. The results showed that ChASO-FOPID and ASO-FOPID outperformed the alternatives’ accuracy and robustness, showcasing their potential for complex engineering control tasks.

Too & Abdullah (2020b) proposed a novel CASO algorithm, integrating twelve chaotic maps into the ASO framework to enhance parameter tuning and convergence efficiency. Tested on twenty UCI benchmark datasets, CASO demonstrated improved performance in feature selection compared to other metaheuristic algorithms. Notably, the logistic-tent map significantly boosted CASO’s effectiveness, optimizing feature selection, improving prediction accuracy, and reducing the number of features required.

Irudayaraj et al. (2022) introduced an enhanced chaotic-based ASO (IASO) algorithm that integrates one-dimensional chaotic maps (logistic, sine, and tent) to enhance exploration and exploitation capabilities. The IASO was validated against classical benchmarks, showing superior performance over standard ASO. It was then applied to optimize a fractional-order PID controller in a multi-source HRES. The IASO demonstrated significant improvements in system performance, including reduced control effort, steady-state error, peak undershoot, and settling time.

Yang, Cai & Zhou (2022) developed an advanced image illumination correction and color restoration model by combining a VGG19-based convolutional neural network (CNN) with an enhanced ASO algorithm. They utilized chaotic-logistic maps to refine ASO, optimizing the regularization random vector functional link (RRVFL) for better illumination estimation and restoration. The V19-CASO-RRVFL model demonstrated improved accuracy and stability, achieving lower average angular errors in illumination estimation than existing methods. This approach proves effective in enhancing color features for computer vision applications.

Zhang et al. (2023c) introduced a wind power prediction model that uses a logistic chaos-enhanced ASO (LCASO) algorithm to optimize a back-propagation (BP) neural network. This approach addresses wind power’s volatility and intermittency by improving prediction accuracy and efficiency. Compared to models like GA-BP and PSO-BP, the LCASO-BP model showed superior performance, demonstrating the effectiveness of ASO variants in enhancing neural networks for complex forecasting tasks in renewable energy systems.

Musthafa et al. (2023) developed a hybrid machine-learning method for early lung nodule prediction by integrating advanced optimization techniques. The authors used an improved snake swarm optimization with a bat model (ISSO-B) for segmentation and a chaotic ASO (CASO) algorithm for feature selection. A deep neural network classifier (L-DNN) was employed for prediction. Tested on datasets like LIDC-IDRI and FAH-GMU, the CASO-enhanced approach showed significant improvements in accuracy, sensitivity, specificity, and AUC, highlighting the effectiveness of ASO in early cancer detection.

Kavitha (2023) introduced a novel multiple disease prediction (MDP) method using retinal fundus images, combining advanced machine learning techniques. Their approach uses an improved weed optimization algorithm (IWO) for disease area segmentation, the SSA for feature extraction, and the chaotic ASO-based deep learning (CAS-DL) classifier for accurate classification. The MDP-HML technique significantly improves over existing classifiers, achieving up to 32.6% higher accuracy and 15% greater sensitivity, demonstrating its effectiveness in disease prediction from retinal images.

Han et al. (2023) developed an advanced indoor positioning system that combines geomagnetic data, light intensity, and inertial navigation measurements. Their method uses a Tent-ASO-BP model, which improves the traditional Back Propagation algorithm by integrating chaos mapping and ASO. The system builds a dual-resolution fingerprint database through radial basis function interpolation and refines positioning with Pedestrian Dead Reckoning data enhanced by a Biased Kalman Filter–Wavelet Transform. The approach significantly reduces positioning errors and enhances robustness, demonstrating the effectiveness of the ASO-based method in indoor positioning.

Chen et al. (2023a) proposed a new forest biomass estimation model that combines Landsat 8 OLI and Sentinel-1A SAR satellite data. The model, enhanced by the Tent Mapping AASO (Tent_ASO_BP) algorithm, outperformed traditional methods like PLSR, SVR, and RF with an R2 of 0.74, RMSE of 11.54 Mg/ha, and MAE of 9.06 Mg/ha, demonstrating superior accuracy in biomass estimation for large-scale forest assessments.

Zhang, Zhang & Wu (2023) developed an advanced ASO (ASO) variant, enhanced with tent mapping, for predicting the state of health (SOH) of lithium-ion batteries. Combined with a back-propagation neural network incorporating temperature and incremental capacity analysis, this approach achieved highly accurate SOH estimates with an error margin below 1.5.

Mundra, Arya & Gawre (2023) introduced a hybrid chaotic ASO (HCASO) method combined with EEMD and LSTM to improve the accuracy of solar and wind power forecasting. This approach effectively addresses forecasting challenges and significantly outperforms existing methods in key performance metrics.

Mehmood et al. (2024) proposed a novel technique for modeling electro-hydraulic actuator systems (EHAS) using an autoregressive exogenous (ARX) model combined with a chaotic variant of ASO (CASO). By integrating ten chaotic maps, including Chebyshev and Logistic, CASO5 significantly reduced the mean-square error (MSE) in model approximation. The method demonstrated superior performance compared to other ASO variants and recent metaheuristics, proving effective in accurately modeling EHAS systems.

Adaptive ASO

Yang et al. (2020) proposed the Fast ASO (FASO) algorithm to improve MPPT for thermoelectric generation (TEG) systems with heterogeneous temperature differences. FASO addresses the issue of multiple local maximum power points (LMPPs) by adaptively updating Euclidean distance ratios and adjusting the number of neighbors for each atom, thereby accelerating convergence to the global maximum power point (GMPP). Simulations and hardware-in-the-loop experiments demonstrated that FASO provides superior energy output compared to single LMPP-based MPPT methods and other metaheuristic algorithms.

Zeng & Shu (2020) introduced the Fast ASO (FASO) to address the reactive power optimization (RPO) problem in power grids with high renewable energy integration. FASO improves search efficiency and avoids premature convergence by optimizing grid components such as generator voltages, VAR compensator capacities, reactive power outputs, and transformer tap ratios. It minimizes voltage deviation and power loss while evaluating renewable energy regulation capacities. FASO was tested on the IEEE 9-bus and 39-bus systems, showing faster convergence and better stability than other methods.

Venkatanaresh & Kullayamma (2022) proposed a hybrid approach combining deep capsule autoencoders with an adaptive ASO variant for crop type prediction in high-resolution satellite images. The method includes preprocessing with refined Lee filtering (RLF) and feature extraction, followed by dimensionality reduction using a modified binary equilibrium optimizer (MBE). The adaptive ASO optimizes network weights, improving classification accuracy and efficiency. Results from Sentinel-2 and Optical Radar datasets show the approach outperforms existing methods.

Tiwary & Mahapatra (2023) proposed a deep learning-based Automatic Image Captioning (AIC) model to assist visually impaired individuals in identifying food items during online grocery shopping. The model integrates an Adaptive ASO (AAS) algorithm with the Extended Convolutional Atom Neural Network (ECANN), which combines LSTM and CNN architectures. The AAS-enhanced ECANN optimizes caption generation, achieving superior accuracy in grocery datasets and showcasing ASO’s effectiveness in accessibility tools and image understanding applications.

Mutation-based ASO

Chitra & Kumar (2021) developed an advanced cervical cancer detection method using Mutation-based ASO (MASO) to enhance the DenseNet 121 architecture. Their approach addresses issues in Pap smear analysis, such as low accuracy and high computational complexity. By optimizing hyperparameters with MASO, including the number of neurons, learning rate, and batch size, the method achieved 98.5% specificity, 98.38% accuracy, and 99.3% recall, significantly outperforming existing methods in detecting cervical cancer across multiple abnormal classes.

Moslemzadeh et al. (2023) introduced an enhanced ASO (IASO) algorithm for optimal reservoir operation in water resources management. IASO improves global search capabilities by incorporating a mutation strategy and reduces local optima trapping. Compared to the original ASO, IASO achieved notable accuracy improvements, reaching 99.99% for a four-reservoir system and 94.36% for a ten-reservoir system, demonstrating its effectiveness in optimizing reservoir operations.

Chitra, Kumar & Subbulekshmi (2024) conducted a comparative study on automated cervical cancer detection using Pap smear images. The first approach optimized DenseNet 121 with Mutation-based ASO (MASO) for hyperparameter tuning. The second approach used a Sooty Tern-optimized CNN-based long short-term memory (ST-CNN-LSTM) classifier with kernel-weighted fuzzy local information C-means (KWFLICM) clustering for segmentation. Evaluated on the Hervel and SIPaKMeD datasets, the ST-CNN-LSTM achieved 99.80% accuracy, 98.83% sensitivity, and 99% specificity, demonstrating superior performance in cervical cancer screening.

Opposition based learning ASO

Ekinci et al. (2020) developed an opposition-based ASO (OBASO) that integrates opposition-based learning to improve search space exploration. The authors applied OBASO to optimize a PIDD2 controller for an automatic voltage regulator system, achieving better control performance by reducing the Integral of Time-Weighted Squared Error (ITSE) and overshoot. The OBASO-PIDD2 demonstrated superior performance compared to the traditional ASO-PIDD2 and other metaheuristic-based controllers, offering enhanced transient and frequency responses as well as robustness against parameter variations.

Shiva, Basetti & Verma (2022) introduced a quasi-oppositional ASO (QOASO) algorithm to improve automatic generation control (AGC) in deregulated power systems. When applied to single- and two-area test power systems, QOASO enhanced frequency control and AGC performance, showing superior results compared to traditional methods in managing deregulated conditions and optimizing system performance.

Levy-based ASO

Anitha & Aravindhar (2024) developed a hybrid machine learning approach using Levy flight-based ASO (LFB-ASO) for retinal detachment diagnosis. The method optimizes feature selection from the Retinal Fundus Multi-disease Image dataset and integrates SVM, GBM, and RF classifiers, achieving high accuracy (98.75%) and superior performance compared to other methods.

Fractional Calculus-based ASO

Menaga & Revathi (2021) introduced a novel cancer classification method combining deep learning with Fractional-ASO (Fractional-ASO). This approach involves preprocessing data using a log transformation, feature selection with a wrapper method, and training a deep recurrent neural network (Deep RNN) optimized by Fractional-ASO, which integrates fractional calculus with ASO. Evaluated on the colon dataset, the model achieved 92.87% accuracy, 93.48% true negative rate (TNR), and 92.87% true positive rate (TPR), demonstrating significant improvements in classification performance and enhancing deep learning’s potential in cancer detection.

Quantum ASO

Mahapatra, Singh & Kumar (2022) proposed the Quantum ASO with Blockchain-aided Data Transmission (QASO-BDT) scheme to improve relay node selection and secure data transmission in wireless ad-hoc networks (WANETs). The approach addresses collusion attacks, energy consumption, delays, and reduced network lifespan. It involves node registration via a capillary gateway, clustering with an enhanced multi-view model, and optimal relay node selection through QASO. Blockchain ensures secure data transmission. Simulation results show that QASO-BDT improves throughput by 91.5%, reduces energy consumption by 40%, lowers end-to-end delay by 20.6%, and decreases node exhaustion to 1%, enhancing both security and network efficiency.

Laplacian operator-based ASO

Saxena et al. (2024) introduced Laplacian-ASO (L-ASO), a novel variant of ASO for identifying and mitigating harmonics in voltage and current signals. L-ASO employs a Laplacian operator-based position update mechanism to enhance ASO’s performance. This approach significantly improves accuracy and effectiveness in designing harmonic estimators, demonstrating superior results compared to traditional methods. Comprehensive analyses validate the potential of L-ASO in optimizing harmonic estimation tasks.

Crossover-based ASO

Joshi et al. (2024) proposed Fusion-Driven Atom Search Optimization with Crossover (FASO-C) for multi-band image fusion. The FASO-C algorithm integrates a crossover operation to improve exploitation capabilities and accelerate convergence to global optima. By optimizing a multi-objective cost function, FASO-C ensures that the fused image preserves essential details and enhances visual quality. Experiments using both subjective and objective metrics confirmed that FASO-C outperforms existing methods in preserving image characteristics and achieving faster convergence.

Mult-strategy ASO

Sun et al. (2020) introduced Cellular Automata and Lévy Flight-Based ASO (CALFASO), an enhanced ASO algorithm designed to overcome local optima stagnation and premature convergence. CALFASO incorporates the cellular automata structure for improved population diversity, the Lévy flight strategy for enhanced global search, and the adaptive weights for better convergence performance. Evaluated on the CEC2017 benchmark and real-world engineering problems, CALFASO outperformed classical and modern evolutionary algorithms, demonstrating an effective balance between exploration and exploitation in optimization tasks.

Fu et al. (2020) developed a modified ASO algorithm incorporating immunologic mechanisms and reinforcement learning to enhance search speed and precision. The method incorporates a vaccine operator to exploit dominant atoms in the population and a reinforcement learning operator that adaptively adjusts the vaccination probability, enhancing the balance between global exploration and local exploitation. Tested on 21 benchmark functions and applied to permutation flow shop scheduling, this modified ASO outperformed seven other algorithms in accuracy, convergence speed, and robustness, demonstrating superior performance and practical applicability.

Rizk-Allah, Hassanien & Oliva (2020) introduced an improved ASO (IASO) algorithm for optimizing shunt capacitor placement and sizing in radial distribution systems. The IASO integrates the learning strategy based on dynamic individual experience (LBDE) and the random search strategy (RSS) to enhance population diversity and solution quality. This approach aims to reduce line losses, minimize capacitor costs, and improve voltage profiles and system reliability. Tested on various IEEE radial systems, IASO outperformed ASO, PSO, SCA, GWO, and SSA, demonstrating its superior performance through statistical validation.

Sun et al. (2021a) introduced DOLHCLASO, a modified version of the ASO algorithm addressing premature convergence and local optima issues. This variant incorporates dynamic opposite learning to enhance exploration and heterogeneous comprehensive learning to balance exploration and exploitation by splitting the population into subpopulations. Evaluated using CEC2017 benchmark functions and real-world engineering scenarios, DOLHCLASO demonstrated superior performance against other methods.

Barshandeh & Haghzadeh (2021) introduced an enhanced ASO algorithm by integrating chaotic maps and Levy flight to balance exploration and exploitation and improve the convergence speed. The authors combined ASO with the Tree-Seed Algorithm (TSA) to boost exploration inspired by tree growth and seed dispersal. This hybrid method effectively reduces local optima stagnation and was tested on benchmark functions and real-world engineering problems, showing superior performance to other meta-heuristic algorithms.

Sun et al. (2021b) proposed an enhanced ASO algorithm to mitigate premature convergence and improve the exploration-exploitation balance. This version features the nonlinear inertia weight factor, the neighbor learning component, an updated mechanism for handling the best and worst atoms, and the greedy selection strategy. Evaluations using CEC2017 benchmark functions and real-world engineering problems demonstrated its superior performance and effectiveness in complex optimization tasks compared to existing algorithms.

Zhang et al. (2022) developed a solar radiation prediction model combining wavelet transform (WT), complete ensemble empirical mode decomposition with adaptive noise (CEEMDAN), and improved ASO (IASO) with outlier-robust extreme learning machine (ORELM). The approach uses WT for data denoising and CEEMDAN for data decomposition. Incorporating Levy flight and chaos initialization, IASO optimizes the ORELM model’s parameters. This model outperformed existing methods in accuracy and robustness, demonstrating its effectiveness for solar radiation forecasting in clean energy applications.

Mohapatra & Patnaik (2022) developed ESAASO to improve epileptic seizure prediction. ESAASO addresses issues in standard ASO by incorporating inertia weight, Levy flight, and ranking strategies. Applied to the CHB-MIT scalp EEG database, with features extracted using TQWT and selected via a genetic algorithm (GA), ESAASO achieved high accuracy in seizure detection, surpassing conventional ASO and other optimization algorithms when used with the LSSVM classifier.

Hua et al. (2022) introduced a wind speed prediction method that combines extreme learning machine (ELM), improved ASO (IASO), partial least squares (PLS), and variational mode decomposition (VMD). IASO, enhanced with simulated annealing and Latin hypercube sampling, optimizes the ELM model for better prediction accuracy. The approach addresses wind speed uncertainty by decomposing wind speed data with VMD and feature extraction using PLS. Tested with data from the Sotavento Galicia wind farm, it outperformed existing short-term wind speed prediction models.

Bi & Zhang (2023) introduced GNUASO, an improved ASO algorithm designed to overcome local optima and premature convergence. GNUASO integrates three key strategies: the global topology with a secant factor for better information exchange, non-linear inertia weights for a balanced exploration and exploitation, and update learning strategy to escape local optima. Evaluations on benchmark test suites and real-world optimization problems demonstrated that GNUASO outperforms existing algorithms, highlighting its effectiveness in complex optimization tasks.

Lu et al. (2023) developed a novel age estimation technique combining Gabor feature fusion with an enhanced ASO algorithm for feature selection. The authors introduced Chaotic Improved Atom Search Optimization with Simulated Annealing (CIASO-SA), which integrates simulated annealing and chaos to speed up convergence and improve accuracy. Applied to the Adience dataset, this method achieved up to 60.4% accuracy in age classification and 85.9% in 1-off accuracy at a resolution of 48×48, outperforming existing techniques and showing promise for mobile applications.

Xiong et al. (2023) introduced a soft measurement model for wastewater treatment that integrates Online Sequential Outlier Robust Extreme Learning Machine (OSORELM), Enhanced ASO algorithm (EASO), and Random Forest (RF). RF identifies vital variables related to effluent parameters, while EASO optimizes OSORELM’s hyperparameters using dynamic perturbation and opposition-based learning. The model demonstrated improved accuracy and stability in predicting ammonia nitrogen and biological oxygen demand, even under varying noise conditions, outperforming existing methods.

Hybridized versions of atom search optimization

A hybrid algorithm combines multiple algorithms to seek optimal solutions for a given problem (Bouaouda & Sayouti, 2022). Researchers across various disciplines worldwide have increasingly recognized ASO. Consequently, diverse applications and hybrid extensions of the basic ASO algorithm have been developed. Hybridization with Particle Swarm Optimization

Zhao et al. (2021) developed the AASOPSO algorithm, which combines Atom Search ASO with PSO to address ASO’s issues with local optima and search efficiency. AASOPSO enhances the exploitation phase with PSO’s cognitive components and uses adaptive acceleration coefficients to balance exploration and exploitation. Additionally, it adjusts each atom’s force constant based on fitness feedback. The algorithm outperformed other optimizers in benchmark tests and in optimizing the no-load PID design of a hydro-turbine governor, demonstrating superior global search performance.

Wang et al. (2022b) proposed the Improved ASO (IASO) to overcome the limitations of the original ASO, such as local optima and lower accuracy. By integrating the velocity update mechanism from PSO, IASO achieved faster convergence and higher precision across 23 benchmark functions. Applied to enhance the maximum likelihood (ML) estimator for the direction of arrival (DOA), IASO demonstrated superior performance in convergence speed, iteration count, and lower root mean square error (RMSE) compared to ASO, SCA, GA, and PSO, showing promise for improving accuracy and reducing complexity in multidimensional optimization tasks.

Izci, Ekinci & Hussien (2023) developed the hybrid Atom Search Particle Swarm Optimization (h-ASPSO) algorithm, merging ASO and PSO to improve optimization by balancing exploration and exploitation. h-ASPSO demonstrated enhanced search efficiency and time-domain performance in complex tasks, such as designing PID controllers and optimizing wind turbine systems. It outperformed the original ASO in convergence speed and solution quality and competed well with other advanced methods while maintaining computational efficiency.

Al Sumarmad et al. (2023) proposed a coordinated power management strategy for battery energy storage (BES) in renewable energy (RE) integrated DC microgrids (MGs) to address power and voltage imbalances from intermittent RE sources. Their system employs a hybrid optimization technique, combining ASO and PSO, to optimize FOPI controllers. This approach enhances the DC network’s control response and voltage regulation, improving BES performance for stable and efficient microgrid operation. Hybridization with simulated annealing

Eker et al. (2021) developed the hybrid Atom Search Optimization (hASO-SA) algorithm by integrating simulated annealing (SA) with ASO. This approach enhances ASO’s search capabilities for nonlinear and linear optimization tasks. Tested on various benchmark functions and applied to practical problems like multilayer perceptron training and PID controller design, hASO-SA outperformed other metaheuristic methods and SA-based hybrids in accuracy and robustness. It showed notable improvements in transient and frequency responses, highlighting its effectiveness across various optimization problems.

Izci (2022) proposed an improved hybrid ASO algorithm by incorporating SA. This variant was designed to optimize power system stabilizers in a single-machine infinite-bus power system. It demonstrated superior performance on benchmark functions compared to Genetic Algorithm, Particle Swarm Optimization, and the original ASO. Additionally, it effectively optimized a power system stabilizer damping controller in real-world applications, outperforming methods such as the Sine-Cosine Algorithm (SCA) and Symbiotic Organisms Search. Hybridization with Sine Cosine Algorithm

Abd Elaziz et al. (2019) developed a hybrid algorithm named ASOSCA, combining ASO with the SCA for automatic clustering. This method optimizes centroids for complex tasks like image clustering and bioinformatics. By integrating SCA, ASOSCA enhances clustering performance and outperforms other metaheuristics on multiple datasets using various cluster validity indexes.

Singh & Kaushik (2021b) developed ASSCA for multi-focus image fusion. This hybrid approach combines ASO with the Sine Cosine Algorithm to optimize the fusion factor in a discrete wavelet transform (DWT) framework. By using Renyi entropy to weigh source images, ASSCA enhances fusion accuracy. The method demonstrated superior performance in metrics like mutual information, peak signal-to-noise ratio, and correlation coefficient, outperforming traditional image fusion techniques.

In other work, Singh & Kaushik (2021a) introduced WeAbDeepCNN, an advanced image fusion technique that tackles contrast reduction, block artifacts, and artificial edges. This method combines a weighted average model with a deep convolutional neural network (DCNN) optimized using the Atom Search Sine Cosine Algorithm (ASSCA). The technique achieves high mutual information, peak signal-to-noise ratio, and low root mean square error, demonstrating superior image clarity and quality compared to existing methods.

Furthermore, Singh & Kaushik (2022) developed a multimodal medical image fusion technique using the Atom Search Sine Cosine Algorithm (ASSCA) with a DCNN. The method integrates Dual-Tree Complex Wavelet Transform (DTCWT) for sub-band decomposition and employs a Renyi entropy-based weighted fusion model. Guided by ASSCA-DCNN, the technique achieved significant improvements in image quality, as demonstrated by higher peak signal-to-noise ratio (PSNR), mutual information (MI), and lower root mean square error (RMSE), highlighting its potential in advancing multimodal medical image fusion.

Zhou et al. (2023) developed the ASCASO algorithm, a hybrid version of the ASO algorithm for better PV model parameter estimation. ASCASO integrates an anti-sine-cosine mechanism, combining sine-cosine algorithm principles with a mutation strategy from linear population size reduction adaptive differential evolution (LSHADE). This approach improves population diversity and communication through arcsine and arccosine functions. Applied to various PV models, ASCASO outperformed nine other algorithms in accuracy and reliability, demonstrating its effectiveness in optimizing PV model parameters. Hybridization with Squirrel Search Algorithm

Borhade & Nagmode (2020) introduced a Modified ASO (MASO) algorithm with a DRNN for epileptic seizure prediction. MASO uses EEG signals to optimize feature selection and DRNN training, enhancing accuracy and reducing computation time. The MASO, which integrates aspects of the Squirrel Search Algorithm and ASO, was tested on the CHB-MIT Scalp EEG dataset, achieving high performance with notable accuracy, sensitivity, and specificity. This approach offers fast convergence, low complexity, and robustness, making it effective for early seizure prediction.

Borhade et al. (2023) developed an Adaptive Exponential Squirrel Atom Search Optimization (Adaptive Exp-SASO) based DRNN for epileptic seizure prediction. This method improves prediction accuracy by using Gaussian filters to clean EEG signals, applying Fuzzy Information Gain (FIG) for feature selection, and optimizing DRNN weights with Adaptive Exp-SASO. The approach achieved 97.87% accuracy, 97.85% sensitivity, and 98.88% specificity, demonstrating strong performance in seizure prediction. Hybridization with Whale Optimization Algorithm

Halhalli, Sugave & Jagdale (2021) introduced a secure routing protocol for mobile ad hoc networks (MANETs) using the Atom Whale Optimization Algorithm (AWOA). This hybrid approach combines ASO and WOA to optimize route selection based on trust factors like encounter rate and cooperation frequency. By integrating these trust metrics into its fitness function, AWOA enhances routing security and performance. The protocol showed significant improvements over existing methods, with reduced end-to-end delay, higher packet delivery ratio, and increased throughput, making it more reliable and efficient for dynamic MANETs.

Singh & Kumar (2024) introduced CWASO-based Deep Stacked Autoencoder (Deep SAE) for crowd behavior recognition. This model uses a combination of ASO and Chaotic Whale Optimization (CWOA) to optimize feature extraction and selection, including statistical and visual features. Evaluated on two datasets, it achieved high precision (96.826%), sensitivity (96.790%), and specificity (99.395%) on dataset 1, demonstrating its effectiveness in crowd behavior recognition. Hybridization with Equilibrium Optimizer

Chattopadhyay et al. (2023) introduced the CEOAS algorithm, a hybrid feature selection method combining a Clustering-based Equilibrium Optimizer (EO) with ASO. This approach improves feature reduction and classification accuracy in high-dimensional machine-learning tasks. Applied to speech emotion recognition, CEOAS effectively extracted features like LPC and LPCC, achieving up to 98.72% recognition accuracy on datasets such as SAVEE, EmoDB, RAVDESS, and IEMOCAP, demonstrating ASO’s effectiveness in optimizing data processing for better machine learning results.

Maazalahi & Hosseini (2024) proposed a two-stage hybrid method for attack detection that combines ASO and Equilibrium Optimization (EO) for feature selection with K-means clustering and the Firefly Algorithm (FA) for classification. This approach, termed ASO-EO-FA-K-means, was tested on NSL-KDD, UNSW_NB15, and KDD_CUP99 datasets, achieving high accuracy (0.998, 0.995, and 0.995) and low error rates, outperforming other methods like PSO and Genetic Algorithms, with notable results on the NSL-KDD dataset. Hybridization with Multi-verse Optimizer

Maheswari, Siva & Priya (2023) introduced the Optimal Cluster-based Intrusion Detection System (OC-IDS), which uses a hybrid Multi-verse and Chaotic ASO (MCA) for data pre-processing and Chaotic Manta-ray Foraging Optimization (CMFO) for clustering. The system achieved 95.01% accuracy on datasets like KDD Cup’99, demonstrating the effectiveness of the hybrid method in cloud security.

Pitchandi, Nivaashini & Grace (2024) proposed an advanced IDS model that combines deep learning with optimization techniques, including Multi-verse-based Chaotic ASO (MCA) for preprocessing, PCA, and Chaotic Manta-ray Foraging Optimization (CMRFO) for feature selection. The ensemble model integrates SVM with CNN, LSTM, and GRU, achieving superior accuracy, precision, recall, and F-measure compared to traditional methods, as demonstrated with the Aegean Wi-Fi Intrusion Dataset (AWID). Hybridization with Grey Wolf Optimizer

Bhagat, Ramesh & Patil (2019) proposed a hybrid method for automatic epileptic seizure diagnosis using the Atom Grey Wolf Optimization (AGWO) algorithm, which combines ASO with GWO. This hybrid approach was designed to enhance the training efficiency of a deep-stacked autoencoder for classifying EEG signals. By leveraging the strengths of both ASO and GWO, the AGWO achieved impressive performance, including 94.103% accuracy, 91.98% sensitivity, and 97.344% specificity, demonstrating its potential for clinical seizure detection. Hybridization with Water Wave Optimization

Dabbu et al. (2022) developed a big data classification approach using a Spark architecture and the Water Atom Search Optimization (WASO) algorithm, which combines Water Wave Optimization (WWO) and ASO algorithm. The process starts with data partitioning via feature selection using Renyi entropy and Black Hole Entropy Fuzzy Clustering (BHEFC). These features are processed by a Deep Recurrent Neural Network (DeepRNN) optimized with WASO. The approach showed significant improvements in classification performance, with higher specificity, accuracy, sensitivity, and precision compared to methods like NN and Support SVM, proving effective for practical big data applications. Hybridization with Rider Optimization Algorithm

Shyjith, Maheswaran & Reshma (2021) proposed the Rider-ASO to improve secure and efficient multipath routing in WSNs. Combining ASO with the Rider Optimization Algorithm (ROA), Rider-ASO optimizes cluster head selection and secure routing. The algorithm achieved notable results: minimal delay of 0.009 s, maximum average residual energy of 0.5494 J, a packet delivery ratio of 97.82%, and a throughput rate of 96.07%, outperforming existing methods. Hybridization with Sunflower Optimization

Jadhav & Joshi (2021) introduced the Atom Search Sunflower Optimization (ASSFO) algorithm to enhance trust-based routing in IoT networks, addressing vulnerabilities like sinkhole attacks. By combining Sunflower Optimization (SFO) with the ASO algorithm, ASSFO leverages multiple trust factors to improve network security and performance. The method surpassed earlier approaches, achieving superior throughput, energy efficiency, trust, and packet delivery ratio (PDR) results. Hybridization with Bird Swarm Algorithm

Alhassan & Zainon (2022) introduced the Atom Taylor BSA-based DBN to enhance medical data classification. This hybrid method combines ASO with the Taylor Bird Swarm Algorithm (Taylor BSA) to optimize feature selection for a Deep Belief Network (DBN) classifier. The approach addresses missing values and redundant attributes, improving classification accuracy. Experimental results, especially using the Hungarian dataset, showed significant gains in specificity, sensitivity, and overall accuracy. Hybridization with Emperor Penguin Optimization

Angel & Jaya (2022) introduced a Hybrid Emperor Penguin Optimization (EPO) algorithm combined with the ASO algorithm to improve Wireless Sensor Networks (WSNs). This hybrid approach enhances EPO’s updating function, focusing on load balancing, security, and energy efficiency. Validated with the NS2 simulator, the method outperformed conventional EPO and ASO in metrics such as delivery ratio, network lifetime, energy consumption, throughput, and delay, providing a robust solution for optimizing WSN performance. Hybridization with Tree-Seed Algorithm

Almajed et al. (2023) developed a forecasting method for non-fungible token (NFT) prices using adaptive improved convolutional neural networks (AICNN) combined with the Tree-seed Chaotic ASO (TSC-ASO) algorithm. This approach incorporates Web3 Blockchain technology to enhance prediction accuracy and integrity. TSC-ASO refines model parameters to improve forecasting reliability. The method outperformed existing techniques, providing more accurate and insightful NFT price predictions in the digital asset market. Hybridization with Aquila Optimizer

Subramaniam & Singh (2023) developed a novel approach for allocating plug-in hybrid electric vehicle (PHEV) charging stations in India, using the hybrid Atom Search Woven Aquila Optimization Algorithm (AT-AQ). This method combines ASO with the Aquila Optimizer (AO) to enhance convergence speed and avoid local optima. The AT-AQ algorithm optimizes the placement of charging stations in a renewable energy system, balancing cost and operational efficiency. The model demonstrated superior performance to existing methods, achieving a lower cost function and providing a practical solution for effective charging station distribution. Hybridization with Marine Predator Algorithm

Edison Prabhu (2023) introduced a machine-learning model for predicting systemic lupus erythematosus (SLE) symptoms, utilizing IoT data to enhance accuracy. The model integrates a hybrid Marine Predator Algorithm (MPA) and ASO with ANNs. Trained on the Gene Expression Omnibus dataset and validated with patient data, the hybrid approach achieved about 99.70% accuracy, outperforming existing models. This highlights the effectiveness of combining ASO with other optimization techniques for improved disease prediction and management. Hybridization with Dolphin Echolocation Algorithm

Deore & Bhosale (2023) proposed the Adaptive DASO to improve intrusion detection systems (IDS). This novel method combines Dolphin Echolocation algorithm with ASO to address the issue of insufficient standard samples. It optimizes feature selection using mutual information and employs a DRNN for intrusion detection. The DRNN, trained with Adaptive DASO, enhances classification accuracy, distinguishing between normal and anomalous network behavior. The approach achieved high performance in specificity, accuracy, and sensitivity, offering a robust solution for IDS. Hybridization with Electromagnetic Force Optimization

Dohare & Singh (2020) developed the energy-balanced integrated ASO and EFO (iASEF) scheme to address rapid energy depletion in IoT sensor networks. The approach integrates the ASO algorithm for selecting effective cluster heads and Electromagnetic Force Optimization (EFO) to determine optimal routing paths. The iASEF scheme uses a linear programming model for clustering, considering node degree, intra-cluster distance, residual energy, and inter-cluster distance. Simulation results showed that iASEF outperforms existing algorithms, significantly improving energy efficiency in sensor networks. Hybridization with Henry Gas Solubility Optimization

Navin et al. (2023) developed a clinical decision support system for diagnosing congenital heart failure by integrating a robust feature selection mechanism. The authors used a hybrid approach combining HGSO and ASO to enhance diagnostic accuracy. ASO was applied in the wrapper-based feature selection process to optimize classification performance, measured by accuracy and G-mean. This innovative use of ASO highlights its effectiveness in healthcare, improving the precision of machine learning models for early detection of cardiovascular conditions. Hybridization with Harris Hawk Optimization

Diab et al. (2020) developed a hybrid control algorithm for Cascaded Modular Multilevel Converters (MMC), integrating HHO and ASO. This approach targets issues such as circulating currents and capacitor stress. The hybrid controller enhances capacitor voltage balancing, reduces harmonic distortion, and minimizes switching frequency. It was tested with an MMC-static synchronous compensator (STATCOM), showing effective stability under different load conditions. The method demonstrates robust performance in controlling MMC systems.

Multi-objective versions of atom search optimization

Previous studies have demonstrated the effectiveness of the ASO algorithm in solving single-objective optimization problems with continuous search spaces. This success has motivated researchers to explore its potential for addressing multi-objective optimization challenges. Junsittiwate, Srinophakun & Sukpancharoen (2022) applied a multi-objective ASO algorithm to optimize palm empty fruit bunch (PEFB) pyrolysis for bio-oil production. Using Aspen Plus simulations and central composite design (CCD), the study evaluated biodiesel yield, CO2 emissions, and utility costs. ASO was employed to maximize yield while minimizing emissions and costs. Compared to CCD, ASO delivered superior results, with a 7.01% increase in biodiesel yield, a 1.33% reduction in CO2 emissions, and a 5.03% reduction in utility costs, showcasing ASO’s efficiency in process optimization.

Asna et al. (2022) introduced a multi-objective binary ASO (MO-QASO), enhanced with quantum operations for optimizing the placement of EV fast-charging stations. The method integrates non-dominated sorting and Pareto optimization to address multiple objectives, such as minimizing grid power loss, station costs, and EV travel time. The quantum-enhanced binarization improves convergence, and the algorithm outperforms traditional binary ASO approaches, demonstrating superior performance in convergence and solution diversity metrics, closely aligning with exhaustive search methods.

Applications of the atom search algorithm

Over recent years, ASO has been applied to various problems, demonstrating its effectiveness across various domains. Due to the multidisciplinary nature of these applications, categorizing them can be complex. Nevertheless, this study aims to organize these applications into specific fields. Table 3 summarizes notable ASO applications in different areas, and the following subsections provide a detailed overview of ASO’s use in these fields.

Table 3 Summary of applications of ASO across various domains.

Domain	Publication	
Engineering	Zhao, Wang & Zhang (2019b, 2019a), Agwa, El-Fergany & Sarhan (2019), Abde-Rahim, Shaaban & Raglend (2019), Kamel et al. (2019), Irudayaraj et al. (2020), Khokhar, Dahiya & Parmar (2020), Gour et al. (2020), Deb et al. (2020), Chaabene & Nehdi (2020), Chandan Kumar Shiva, Kumar & Kumar (2020), Khokhar, Dahiya & Parmar (2021), Amarendra, Srinivas & Rao (2021), Kumar & Sinha (2021), Al-Akayshee, Kuznetsov & Sultan (2021), Khonturaev et al. (2021), Rao, Srikanth Goud & Rami Reddy (2021), Reddy et al. (2021), Khatir et al. (2021), Ranjitha, Ponnurangam & Rajapandiyan (2021), Gupta et al. (2021), Duan et al. (2022a), Bahr et al. (2022), Bagade & Rampure (2022), Mishra et al. (2022), Tahiliani & Gupta (2022), Bie et al. (2022), Jiang et al. (2020), Dey, Banerjee & Dey (2022), Ekinci et al. (2023), Goud et al. (2023), Kala et al. (2023), Isabella, Umamaheswari & Marimuthu, 2023, Pavani et al. (2023), Sepperumal et al. (2023), Chen et al. (2023b), Sekar et al. (2023), Wang et al. (2023), Osama Abed El-Raouf et al. (2023), Kumar & Sikander (2023), Hao et al. (2023), Saadi et al. (2023), Güven & Mengi (2023), Mohamed, Hasanien & Ebrahim (2024)	
Healthcare and medical applications	Surenthiran, Rajalakshmi & Sujatha (2021), Toğaçar (2022), Marzouk et al. (2022), Vadlamudi et al. (2023), Ali et al. (2024)	
Internet of Things and communication	Almagboul et al. (2019), Nath, Som & Negi (2020), Elkamchouchi et al. (2022), Korkmaz & Akgüngör (2022), Al Duhayyim et al. (2022), Srinivasiah, Ranganathasharma & Ramanna (2023)	
Clustering and data mining	Karadeniz, Çelik & Başaran (2023), Mao et al. (2023), Özbay, Özbay & Gharehchopogh (2024)	
Environmental modeling	Cao et al. (2022), Ghimire et al. (2022), Kosasaeng et al. (2022), Ann et al. (2022), Li et al. (2023), Azma et al. (2023), Ghosh et al. (2024)	
Security	Saba et al. (2021), Lavanya & Kavitha (2021), Silivery, Rao & Kumar (2023).	

Engineering

During its development, Zhao, Wang & Zhang (2019b) tested the performance of ASO by applying it to hydrogeologic parameter estimation and dispersion coefficient estimation (Zhao, Wang & Zhang, 2019a) problems. ASO demonstrated superior performance to other popular algorithms, such as PSO and GA, highlighting its effectiveness and competitiveness in solving real-world optimization problems.

Abde-Rahim, Shaaban & Raglend (2019) utilized ASO to address the optimal power flow (OPF) problem. The OPF-ASO method was tested on the IEEE 30-bus system and showed superior performance to other techniques, underscoring ASO’s robustness and effectiveness in solving complex power flow optimization issues.

Kamel et al. (2019) applied the ASO algorithm to solve the optimal distribution system reconfiguration problem, focusing on minimizing power losses in electrical networks. ASO adjusted the network topology to find configurations that reduce active power losses. The study demonstrated that ASO outperformed other optimization methods, proving its effectiveness in enhancing power distribution systems.

Irudayaraj et al. (2020) employed the ASO algorithm to optimize an FOPID controller for Automatic Load Frequency Control in Hybrid Renewable Energy Systems (HRES). The ASO-tuned FOPID controller improved performance by minimizing the Integral Time Absolute Error, particularly addressing frequency issues arising from intermittent renewable sources and Plug-in Electric Vehicles (EVs). Simulations in MATLAB/Simulink and hardware-in-the-loop tests validated the controller’s enhanced transient and steady-state responses, demonstrating its robustness and real-time applicability.

Khokhar, Dahiya & Parmar (2020) utilized the ASO algorithm for load frequency regulation in HRES by optimizing an integer-order PID controller with ASO. The ASO-based controller exhibited superior performance compared to other approaches in minimizing frequency variations caused by load and wind power fluctuations. The effectiveness of the controller was validated using Bode diagrams and eigenvalue analysis, confirming its robustness in enhancing frequency control for HRES.

Diab et al. (2020) developed a hybrid control algorithm for Cascaded Modular Multilevel Converters (MMC), integrating the HHO and ASO algorithms. This approach addresses issues such as circulating currents and capacitor stress. The hybrid controller enhances capacitor voltage balancing, reduces harmonic distortion, and minimizes switching frequency. It was tested with an MMC-static synchronous compensator (STATCOM), demonstrating adequate stability under different load conditions. The method shows robust performance in controlling MMC systems.

Gour et al. (2020) proposed an optimization framework for improving the placement and sizing of distributed generation (DG) units to minimize power loss in distribution systems. The authors used loss sensitivity factors (LSF) to identify optimal DG locations and applied the ASO algorithm to optimize power loss reduction while adhering to system constraints. Tested on 10-bus and 33-bus radial distribution systems (RDS), the ASO-based framework demonstrated significant improvements in power loss reduction, highlighting ASO’s effectiveness in optimizing complex energy systems.

Deb et al. (2020) proposed using the ASO algorithm to address transmission line congestion in deregulated electricity markets by rescheduling generator outputs. ASO helps optimize real power adjustments to manage congestion effectively. Evaluated on the IEEE 30-bus test system, the method demonstrated improved efficiency and reliability in power transfer, alleviating congestion and enhancing overall system performance.

By integrating the ASO algorithm with ANN, Chaabene & Nehdi (2020) presented a technique for estimating the shear capacity of steel fiber-reinforced concrete (SFRC) beams. ASO optimizes ANN parameters for accurate shear strength predictions. Four classification models (naïve Bayes, SVM, decision tree, KNN) were also used to predict failure modes. ASO-ANN showed the highest accuracy in shear strength prediction, and KNN excelled in failure mode classification. This approach improves SFRC design and retrofitting strategies.

Chandan Kumar Shiva, Kumar & Kumar (2020) designed a dynamic controller for load frequency control (LFC) using a three-degree-of-freedom PID (3DOF-PID) controller optimized with the ASO algorithm. Tested on a four-area power system, the ASO-based 3DOF-PID controller showed superior performance compared to traditional methods, improving settling time, peak variance, and oscillation magnitude. This highlights its effectiveness in enhancing LFC applications.

Khokhar, Dahiya & Parmar (2021) developed an LFC scheme for multi-microgrid systems using electric vehicles (EVs) and a tilt integral derivative (TID) controller optimized by ASO algorithm. This approach improved dynamic performance, reduced oscillations, and enhanced stability under various disturbances, outperforming standard algorithms even with ±30% parameter variations.

Amarendra, Srinivas & Rao (2021) used ASO to optimize the placement of FACTS devices (UPFC, TCSC, IPFC, SVC) to enhance power system security. The ASO-based method was evaluated on IEEE 30, 118, and 300 bus systems using metrics like severity index and voltage stability. ASO demonstrated superior performance in optimizing FACTS device placement compared to conventional algorithms, including DA, WOA, Jaya, FPA, GWO, and JA-FPA.

Kumar & Sinha (2021) proposed a new method for managing power demand and voltage stability in an autonomous microgrid with renewable sources like solar dish-stirling generators. They incorporated battery storage to handle periods of insufficient solar energy and used a classical PI controller for simplicity. The ASO algorithm was employed to optimize power management and voltage stability, showing superior results compared to PSO. Simulations on MATLAB/Simulink validated the effectiveness of the ASO-based approach.

Al-Akayshee, Kuznetsov & Sultan (2021) developed an off-grid hybrid energy system using PV, biomass, and hydroelectric pumped storage to meet power needs in remote Iraq. The author utilized the ASO algorithm to determine the optimal system size and minimize net present cost (NPC), considering real-time load and meteorological data. The ASO-based approach provided an economical and sustainable solution for isolated areas, supporting the development of reliable, clean energy systems in rural regions.

Goud & Rao (2021) introduced an approach to improve power quality (PQ) in HRES using the ASO algorithm and a unified power quality conditioner (UPQC). The authors optimized the UPQC with a FOPID controller designed via ASO for both series and shunt filters. The MATLAB/Simulink simulations showed that the ASO-based method effectively mitigated PQ issues such as voltage sags, swells, and total harmonic distortion (THD), outperforming traditional PI controllers and other optimization methods.

Khonturaev et al. (2021) developed a hybrid approach for optimizing the sizing and placement of distributed generations (DG) in RDS by combining the ASO algorithm with a power loss sensitivity (PLS) index. This approach uses ASO to find optimal DG locations and sizes while the PLS index identifies the best buses for installation. Tested on the IEEE 33-bus RDS, the method improved voltage profiles and reduced power losses significantly, outperforming existing techniques in enhancing grid efficiency and sustainability.

Rao, Srikanth Goud & Rami Reddy (2021) integrated an HRES with grid systems to enhance electricity demand management and reduce environmental impact. The authors used the ASO algorithm to optimize control parameters for a Unified Power Flow Controller (UPQC), addressing issues like voltage sag, swell, and total harmonic distortion (THD). The ASO-optimized fractional-order PID controller outperformed conventional PI controllers, with MATLAB/Simulink simulations confirming its effectiveness in improving power quality in grid-connected HRES.

Reddy et al. (2021) introduced an intelligent control approach for enhancing power quality in grid-connected hybrid power systems with solar, wind, and battery storage. The authors used a Unified Power Quality Conditioner with Active and Reactive Power (UPQC-PQ) combined with an FOPID controller optimized by the ASO algorithm. This method regulates voltage, minimizes power loss, and reduces THD. The ASO-based FOPID controller outperformed traditional controllers and optimization techniques, demonstrating superior performance in various operating modes through MATLAB/Simulink simulations.

Khatir et al. (2021) proposed a damage detection methodology using a flexibility index to locate and quantify damage in complex truss structures. The authors applied SSA and ASO algorithms based on enhanced damage indicators to solve the inverse problem. ASO showed superior convergence performance and lower CPU time compared to SSA. The method was validated on the Guangzhou TV Tower, demonstrating its accuracy and efficiency in predicting damage across different structural floors.

Ranjitha, Ponnurangam & Rajapandiyan (2021) proposed an advanced control strategy for load frequency management in a hybrid power system with wind, solar, and thermal sources. Their method integrates a cascaded PID controller with an ASO-tuned PID controller to improve performance and cost management. The strategy also uses a firefly-based controller for thermal power optimization. This approach demonstrated superior effectiveness to traditional PID and fuzzy logic controllers in simulations, efficiently handling system stability and cost optimization despite renewable energy fluctuations.

Gupta et al. (2021) comprehensively evaluated nine metaheuristic algorithms, including the ASO algorithm, for complex mechanical design problems. The study assessed algorithms such as the Moth-Flame Optimizer (MFO), Multi-Verse Optimizer (MVO), Salp Swarm Algorithm (SSA), and others across eight design challenges. The evaluation focused on solution quality and convergence characteristics, revealing the effectiveness of these algorithms, including ASO, in solving real-world mechanical design issues and highlighting their broad potential for tackling complex optimization tasks.

Duan et al. (2022a) used the ASO algorithm to design gravity-driven drip irrigation systems. The model focuses on minimizing costs while ensuring operational efficiency by optimizing variables such as pipe layout, diameter, emitter-flow rate, and water pressure. Applied to a two-level irrigation project in Xinjiang, ASO effectively reduced the cost per unit area and improved efficiency. The study showed that cost efficiency improved with greater emitter and pipe spacing but decreased with higher emitter-flow rates, highlighting ASO’s effectiveness in optimizing irrigation systems for both cost and performance.

Bahr et al. (2022) proposed an advanced method for managing the dynamic performance and stability of an isolated hybrid micro-grid (HMG) system using solar, wind, and biomass energy sources. They optimized frequency and power control with PID and PIDA controllers, fine-tuning them with the ASO algorithm to minimize the integral of time-weighted absolute error (ITAE). MATLAB/Simulink simulations showed that the ASO-optimized PIDA controller outperformed the conventional PID controller, offering superior frequency stability and reliability under various system uncertainties.

Bagade & Rampure (2022) developed an optimization strategy for enhancing the efficiency of a micro-combined cooling, heating, and power generation (MCCHP) system using the ASO algorithm. The approach optimizes variables in a single-stage, double-effect LiBr-H2O absorption chiller, integrating components such as a diesel engine, heat exchanger, and boiler. The ASO-optimized system achieved a 16 kW cooling capacity with a 98% improvement in temperature reduction and enhanced cost-effectiveness. This optimization significantly benefits energy efficiency and economic viability in hospitals and food production.

Mishra et al. (2022) introduced an ASO-based multistage PID (M-PID) controller for frequency control in multi-microgrid systems with variable renewable energy sources. Their system includes PV panels, wind turbines, diesel generators, and energy storage devices. The ASO-based M-PID controller was compared to GA and PSO, showing superior performance in frequency management. Validation through OPAL-RT simulations demonstrated its effectiveness in handling stochastic conditions and improving control performance.

Tahiliani & Gupta (2022) developed an ASO-based method for optimizing the placement of distributed generators (DG) and distribution static compensators (D-STATCOM) in unbalanced radial distribution systems. Their approach aims to enhance voltage profiles, reduce power losses, and assess cost savings. ASO is used to determine optimal DG and D-STATCOM placement under varying conditions, improving overall power distribution system efficiency. The method’s effectiveness was validated using MATLAB simulations on a 25-bus network.

Bie et al. (2022) introduced a fault diagnosis method for gearboxes that combines the ASO algorithm with resonance-based sparse signal decomposition (RSSD) and support vector machine (SVM) techniques. ASO optimizes the quality factor in RSSD to enhance fault detection by decomposing vibration signals and calculating entropy measures. SVM is then used for fault pattern recognition. The approach demonstrated improved accuracy in identifying gearbox faults in the presence of interference, as shown by simulations and experimental results.

Jiang et al. (2020) developed a hybrid forecasting framework combining Fuzzy Time Series (FTS) with the ASO algorithm to improve forecasting of inbound tourism demand in China, especially with small sample sizes. The framework uses FTS for better information recognition and ASO for optimizing FTS parameters, enhancing forecasting accuracy. Validation through comparative experiments showed that this approach outperforms traditional models, providing reliable forecasts and aiding better planning for the tourism industry.

Dey, Banerjee & Dey (2022) utilized the ASO algorithm to optimize PID controllers for an unstable Magnetic Ball Levitation System (Maglev). The study focused on various PID types, including integer order PID (IOPID), IOPID with a derivative filter (IOPID-F), fractional order PID (FOPID), and FOPID with a fractional order filter (FOPID-F). ASO was employed to fine-tune controller parameters for enhanced stability and transient performance. Results showed that the FOPID-F controller achieved the best stability, input tracking, and robustness, demonstrating ASO’s effectiveness in tackling complex engineering challenges.

Ekinci et al. (2023) proposed using the ASO algorithm for designing digital infinite impulse response (IIR) filters in digital signal processing. The study compared ASO with other metaheuristic algorithms, such as moth flame optimization and gravitational search algorithms. As evidenced by pole-zero diagrams, ASO demonstrated the highest improvement in filter design performance and superior stability. This highlights ASO’s effectiveness and robustness in optimizing IIR filters, outperforming other methods.

Goud et al. (2023) conducted a comprehensive analysis of optimization techniques, including the ASO algorithm, to enhance power quality in electrical systems. The study reviews various metaheuristics like Whale Optimization Algorithm (WOA), Black Widow Optimization (BWO), Gravitation Search Algorithm (GSA), and ASO, emphasizing their effectiveness in managing complex nonlinear systems. The article demonstrates the significant potential of these algorithms, especially in optimizing power quality when integrating renewable energy and storage systems into power networks.

Kala et al. (2023) proposed an ASO-based method for selective harmonic elimination (SHE) in multilevel inverters (MLI). This approach uses ASO to optimize firing angles in an 11-level MLI system, effectively reducing low-order harmonics while maintaining fundamental voltage control. Compared to other metaheuristic techniques such as bee algorithm, ICA, firefly optimizer, PSO, and TLBO, the ASO method demonstrated superior performance in reducing total harmonic distortion (THD) and harmonics, with successful results from simulations and experiments on a photovoltaic-based RCC MLI.

Isabella, Umamaheswari & Marimuthu (2023) introduced a high-gain BOOST-CUK (BOCUK) DC-DC converter for switched mode power supply (SMPS) applications, combining Boost and Cuk converters for better voltage gain and efficiency. The authors optimized the performance using an ASO-tuned PI controller within a dual-loop control system. This approach improved line and load fluctuations regulation, demonstrating ASO’s potential in refining control systems for efficient and stable power conversion with reduced harmonic distortion and improved power factor.

Pavani et al. (2023) applied the ASO algorithm to synthesize linear and planar antenna arrays, optimizing symmetric and asymmetric linear arrays and two-ring concentric circular arrays with non-uniform amplitude coefficients. ASO outperformed other algorithms, such as the Flower Pollination Algorithm (FPA) and Firefly Algorithm (FA) in pattern synthesis. This demonstrates ASO’s effectiveness in solving complex electromagnetic optimization problems, making it a valuable tool in antenna design.

Sepperumal et al. (2023) proposed an ASO-tuned FOPI controller for enhancing a DC–DC SEPIC converter used in power factor correction and voltage regulation. The ASO algorithm optimized the controller’s parameters in voltage and current control loops, resulting in superior dynamic response, efficiency, and power factor compared to a PSO-optimized FOPI controller. The ASO-tuned controller also demonstrated improved stability and voltage regulation under varying loads, showcasing ASO’s effectiveness in optimizing power electronics control systems.

Chen et al. (2023b) proposed an ASO-Back Propagation (ASO-BP) neural network to predict wear rates of copper-based powder metallurgy brake pads. By optimizing the weights and thresholds of a Back Propagation neural network with ASO, the model achieved 97.3% prediction accuracy. The study highlighted the significant impact of braking speed and pressure on performance, improving the reliability of brake systems in high-speed trains.

Sekar et al. (2023) developed an advanced method to enhance power quality and reactive power compensation in grid-connected PV systems. The authors used the ASO algorithm for MPPT to optimize energy extraction from PV panels. The approach includes a non-isolated high voltage gain DC-DC converter and a Coyote Optimized Converter Control (COCC) for precise switching. A residual attention echo state reactive controller (RaERC) also improves control signals for a nine-level inverter. Simulations validated significant improvements in power quality and efficiency.

Wang et al. (2023) introduced a planetary gear fault diagnosis method for wind turbines using a digital twin model integrated with ASO algorithm and SVM techniques. The system combines real-time and virtual data to improve fault detection, overcoming the limitations of traditional methods. Using empirical mode decomposition (EMD) with ASO-SVM, the approach enhances diagnostic accuracy and operational efficiency. Implemented on Unity3D, it offers advanced real-time monitoring and early-warning capabilities for wind turbine maintenance.

Osama Abed El-Raouf et al. (2023) proposed enhancing grid-tied renewable energy systems with a unified power flow controller (UPFC) regulated by a FOPID controller. The authors optimized the FOPID parameters using the ASO algorithm. The study showed that the ASO-tuned FOPID controller improved system stability and power quality during disturbances, reduced total harmonic distortion, and supported maximum power point tracking for photovoltaic cells and wind turbines.

Kumar & Sikander (2023) introduced a method for creating reduced-order models (ROMs) of large-scale linear time-invariant (LTI) systems using the ASO algorithm. The ASO technique simplifies single-input single-output (SISO) systems by minimizing the integral square error (ISE) between the original and reduced-order models. Using molecular dynamics to optimize search variables, the ASO-based ROMs demonstrated superior time and frequency response compared to traditional methods. This approach highlights ASO’s efficiency and accuracy in control engineering applications.

Hao et al. (2023) proposed an optimization model for short-term traffic flow prediction, integrating the ASO algorithm with Extreme Learning Machine (ELM). The ASO-ELM model optimizes ELM performance, addressing the challenges of dynamic and nonlinear traffic flow. Tested with A10 ring road data in Amsterdam, the model improved prediction accuracy, reducing mean absolute percentage error (MAPE) by 9.8% and root mean squared error (RMSE) by 5.9% compared to traditional models like ANN. This approach highlights the effectiveness of combining ASO with ELM for accurate traffic flow forecasting.

Saadi et al. (2023) developed a high-performance control strategy for an interleaved non-isolated DC/DC converter in fuel cell applications, using the ASO algorithm for controller design. The strategy features a two-degree-of-freedom PID outer loop for setpoint tracking and disturbance rejection and a super-twisting integral sliding mode (STISM) inner loop for rapid convergence and reduced chattering. ASO optimizes controller gains for robust and stable operation. Validated through simulations and real-time tests, the ASO-based approach outperforms other methods and dual-loop PID controllers in performance.

Zhang et al. (2023a) introduced SPAA, a sample-wise probabilistic approach combining ASO with ANN to assess the stability of circular shafts in urban infrastructure, considering soil variability. This ASO-ANN model replaces extensive simulations with Latin hypercube sampling and iterative enrichment. Validated by Monte-Carlo simulations and global sensitivity analysis, SPAA offers reliable failure probabilities and sensitivity indices while reducing simulation requirements, outperforming traditional methods in high-dimensional scenarios.

Güven & Mengi (2023) developed a framework using the ASO algorithm to optimize HRES in Turkey. The authors integrated wind, solar, biomass, and fuel cells with excess energy stored as hydrogen. ASO was used to fine-tune variables like wind and solar power and hydrogen storage, achieving the most cost-effective solution and best convergence among the ten algorithms. The approach was validated through MATLAB simulations, demonstrating significant improvements in energy flow management and cost reduction.

Mohamed, Hasanien & Ebrahim (2024) improved PV systems’ performance under partial shading by using the ASO algorithm to optimize global MPPT controllers. The authors integrated ASO with Proportional-Integral, Fractional-Order PI, and Fuzzy Logic Controllers, achieving better tracking accuracy, faster convergence, and higher power efficiency. ASO-based Fuzzy IC MPPT controllers showed reduced settling times and improved stability.

Healthcare and medical applications

Surenthiran, Rajalakshmi & Sujatha (2021) introduced a DBNN optimized with the ASO algorithm for predicting student performance. The DBNN uses a cognitive divergence algorithm with restricted Boltzmann machines (RBMs) to enhance learning rate optimization. The model achieved 90% accuracy and reduced error rates to below 20%. It outperformed previous methods, offering valuable insights for early academic interventions.

Marzouk et al. (2022) developed the ASODCAE-SLR model, which combines the ASO algorithm with a deep convolutional autoencoder (DCAE) for Arabic Sign Language (ASL) recognition. The model addresses the challenge of recognizing complex hand gestures using weighted average filtering for input frame preprocessing and a capsule network (CapsNet) for feature extraction. ASO optimizes hyperparameters to enhance DCAE performance. Tested on the Arabic Sign Language dataset, the ASODCAE-SLR model achieved superior recognition accuracy compared to existing methods, improving communication for individuals with hearing and speech disabilities.

Toğaçar (2022) developed an advanced method for detecting diabetic retinopathy by integrating the ASO algorithm with deep learning techniques. Their approach involves preprocessing fundus images using morphological gradients and segmentation to enhance image quality and extract ocular vessel information. Transfer learning is employed to train CNNs on publicly available datasets, extracting activation sets for diabetic retinopathy severity classes. ASO is then used to select the most influential activation sets, improving classification performance. This method achieved accuracy rates of 99.59% and 99.81% on two separate datasets, significantly enhancing the early detection of diabetic retinopathy.

Vadlamudi et al. (2023) proposed an advanced heart disease detection model using deep learning techniques for ECG signal analysis. The model includes pre-processing with a discrete wavelet transform (DWT) filter, feature extraction through CNN, and classification of heart conditions. Although traditional optimization methods like ASO and GWO were used, the deep learning approach demonstrated superior performance, achieving high sensitivity, accuracy, and recall. This approach proves to be more competitive in heart disease prediction than existing techniques.

Ali et al. (2024) proposed a wrapper-based technique to enhance chest infection detection using X-ray images. The authors extracted deep features with pre-trained models and optimized these features using ten algorithms, including ASO, PFA, HGSO, HHO, MRFO, EO, SMA, GNDO, and MPA. Combined with network selection, the method achieved a high classification accuracy of 97.7% on a chest infection X-ray dataset, significantly improving detection performance and demonstrating strong potential for clinical use.

Internet of Things and communication

Almagboul et al. (2019) developed an ASO-based method for optimizing hybrid analog-digital (HAD) beamforming in 5G cellular networks. This approach uses a partially connected HAD beamformer to address challenges such as high energy consumption and hardware costs. ASO optimizes beamforming weights by adjusting digital amplitudes and analog phase shifts to minimize peak sidelobe levels, accurately steer nulls, and reduce power consumption. Simulation results showed that ASO consistently outperformed other metaheuristic algorithms in sidelobe reduction, null depth, and convergence, demonstrating its effectiveness for large-scale antenna systems in 5G networks.

Nath, Som & Negi (2020) introduced an approach to enhance IoT security by combining the ASO algorithm with Elliptic Curve Cryptography (ECC). The ASO-based ECC optimizes key generation to improve encryption and decryption efficiency. This method addresses security vulnerabilities by selecting optimal keys, leading to a reduced encryption time of 2.7384 s and better key management. Simulation results showed that the ASO-enhanced ECC outperformed traditional methods in encryption time, entropy, and Peak Signal-to-Noise Ratio (PSNR), demonstrating superior system security and efficiency.

Elkamchouchi et al. (2022) proposed the ASO-based Clustering with Encryption Technique for Secure Internet of Drones (ASOCE-SIoD) to enhance security and efficiency in Internet of Drones (IoD) environments. ASO is used to form clusters and designate cluster heads, while an encryption technique encrypts images captured by drones. This approach ensures secure transmission of visual data to ground stations. Experimental results showed that ASOCE-SIoD outperformed existing methods, particularly in clustering efficiency and data security within dynamic IoD environments.

Korkmaz & Akgüngör (2022) introduced a new approach for predicting traffic signal cycle lengths, improving the traditional Webster model, which struggles in high-traffic scenarios. The authors developed predictive models for different traffic conditions using ASO and GOA. Evaluated in exponential, power, and quadratic forms, the ASO-based models outperformed the Webster model and VISTRO optimization, providing more accurate predictions and reducing delays. Statistical analyses confirmed that ASO was superior to GOA in predicting optimal cycle lengths.

Al Duhayyim et al. (2022) developed a smart water quality prediction approach using the ASO algorithm with a fuzzy deep convolutional network (F-DCN) in an IoT environment. By optimizing hyperparameters with ASO, their method improved prediction accuracy for water quality parameters monitored by IoT devices. Simulations showed that the ASO-enhanced F-DCN model significantly outperformed existing techniques, offering a promising solution for water quality monitoring in IoT-based systems.

Srinivasiah, Ranganathasharma & Ramanna (2023) introduced the trust-aware clustering and routing protocol (TCRP) for wireless sensor networks (WSNs), leveraging the ASO algorithm to improve data security and energy efficiency. TCRP enhances node security by selecting secure cluster heads (CHs) and establishing protected routing paths with node authentication. The fitness function considers distance, trust, and energy. TCRP outperformed existing protocols such as SQEER and PFCD-ARP-CH, achieving 99% throughput with 80 nodes and demonstrating superior performance in energy consumption, throughput, and packet delivery ratio.

Clustering and data mining

Karadeniz, Çelik & Başaran (2023) developed a method for distinguishing walnut varieties using image processing and optimization. The authors created a dataset of 1,751 leaf images from 18 walnut species and used a residual block-based convolutional neural network to extract deep features. The ASO algorithm was applied to select the most distinguishing features, which were then classified using a Support Vector Machine (SVM), achieving 87.42% accuracy. This approach highlights the effectiveness of combining ASO with deep learning for precise classification in botanical studies.

Mao et al. (2023) introduced the Fast Mahalanobis Classification System (FMCS) for high-dimensional and imbalanced data in hydraulic system fault diagnosis. FMCS employs a two-stage dimensionality reduction method, combining symmetrical uncertainty with Mahalanobis kernel PCA. The Mahalanobis distance assesses feature importance, and the ASO algorithm improves threshold determination in the Mahalanobis-Taguchi system. The FMCS model outperformed 24 baseline methods, achieving high classification accuracy (0.92), an F1 score (0.94), and a G-mean (0.91), with reduced computation time (4.9045 s), demonstrating its effectiveness in intelligent fault diagnosis for hydraulic systems.

Özbay, Özbay & Gharehchopogh (2024) developed a methodology for mushroom species classification using a dataset of 6,714 images from nine species. The authors applied visualization techniques such as Grad-CAM, LIME, and Heatmap to enhance feature extraction. The features were optimized using the ASO algorithm, significantly reducing the feature map size. Coupled with a residual block-based CNN, the ASO-optimized approach achieved a 95.45% classification accuracy with the KNN classifier. This demonstrates ASO’s effectiveness in feature optimization and highlights the benefit of advanced visualization techniques for mushroom species identification.

Environmental modeling

Cao et al. (2022) developed a short-term PV power forecasting model that combines the ASO algorithm with a backpropagation neural network (BPNN). ASO optimizes the BPNN’s weights and thresholds, while the Pearson correlation coefficient is used to select key meteorological factors, such as irradiance and temperature. The Euclidean distance formula customizes training sets for each instance. Applied to a solar farm in Yunnan, China, the ASO-BPNN model demonstrated improved accuracy in forecasting PV power output, enhancing the operational efficiency and stability of grid-connected PV systems.

A hybrid deep learning model called CSVR was presented by Ghimire et al. (2022), which combines NN and SVR to forecast global solar radiation (GSR). CNN extracts features from time series data, while SVR forecasts daily GSR. The ASO algorithm was used to select key meteorological variables, and HyperOpt optimized the hyperparameters. The CSVR model demonstrated superior predictive accuracy with lower errors than other methods, highlighting ASO’s role in improving feature selection and enhancing renewable energy forecasting.

Duan et al. (2022b) developed an optimization model to minimize total investment in a multilevel gravity drip irrigation system’s pipe network, using the ASO algorithm to enhance pipe layout and diameter selection. The model optimizes the field drip irrigation pipe network (FDIPNS) and then refines the main water distribution system (MWDPNS) based on the FDIPNS results. By addressing topological and hydraulic constraints, the ASO-based method improved the pipe network design in Xinjiang, China, demonstrating its practical benefits for managing irrigation systems in irregularly shaped fields.

Kosasaeng et al. (2022) utilized the ASO algorithm, wind-driven optimization, and GP to optimize reservoir rule curves for a multi-reservoir system in Sakon Nakhon, Thailand, addressing the severe flooding of 2017. Their method aimed to reduce excess water release by optimizing reservoir operations using hydrological data and water demand. The application of ASO demonstrated that managing multiple reservoirs was more effective than single-reservoir systems, enhancing water management efficiency without needing new infrastructure.

Ann et al. (2022) developed a hybrid optimization method combining multi-gene symbolic regression with the ASO algorithm to minimize evapotranspiration (ET) in aquaponic lettuce. By optimizing artificial light properties, the approach improved plant growth and water efficiency in controlled-environment agriculture (CEA). ASO, when used with genetic programming, found the optimal solution, enhancing yield and growth. This highlights ASO’s effectiveness in agricultural optimization and resource management in CEA systems.

Li et al. (2023) introduced a PV reconfiguration method using the ASO algorithm to address hot spot effects from partial shading. Evaluated by mismatch loss, fill factor and standard deviation, their method showed better efficiency, speed, and reliability than other techniques. This demonstrates ASO’s potential for optimizing energy systems and enhancing PV array performance and durability under diverse conditions.

Azma et al. (2023) proposed two hybrid models—BBO-ANN and ASO-ANN—to predict dissolved oxygen (DO) levels in the water. Combining ASO with ANN and BBO, these models used 5 years of data from Rock Creek. The ASO-ANN model achieved a mean absolute percentage error (MAPE) of 2.5170% and a correlation of 0.99135, while the BBO-ANN model had a slightly better MAPE of 2.3848% and a correlation of 0.99186. Both models outperformed existing methods, with the BBO-ANN providing a practical formula for DO prediction.

Ghosh et al. (2024) developed a predictive framework for modeling clean electricity generation in India, focusing on hydro, nuclear, and renewable sources. The authors combined ensemble empirical mode decomposition, density-based clustering, and the ASO algorithm to forecast energy production. The framework effectively modeled hydroelectric generation, particularly during volatile periods like the COVID-19 pandemic, highlighting the role of industrial demand in clean energy growth.

Security

Saba et al. (2021) developed an automated exam proctoring system by integrating the ASO algorithm with a deep learning model called L2-GraftNet, which combines AlexNet and SqueezeNet features. ASO optimized feature extraction, and the fine KNN classifier achieved 93.88% accuracy, demonstrating the system’s effectiveness in automating exam monitoring and reducing manual supervision.

Lavanya & Kavitha (2021) introduced an advanced iris recognition framework to improve biometric identification accuracy. The authors used a combination of SUSAN, GHT, and VJ techniques for eye detection, along with fuzzy Retinex for image enhancement. ASO was applied to optimize the matching and classification process in an FFCNN, achieving a high recognition rate of 99.9%, surpassing other methods.

Silivery, Rao & Kumar (2023) developed an intrusion detection system for DoS and DDoS attacks, using deep convolutional generative adversarial networks (DCGAN) to address class imbalance and ResNet-50 for feature extraction. The authors applied ASO to optimize an AlexNet-based model, achieving high accuracy on the CICIDS2019 and UNSW-NB15 datasets, highlighting ASO’s effectiveness in optimizing deep learning models for cybersecurity.

Open source software of aso

The ASO algorithm has recently garnered significant interest from researchers seeking to apply it across various fields. To ensure comprehensiveness in this work, we provide detailed information and URLs for all available open-source implementations of this promising technique. Since its introduction, the ASO algorithm has been implemented in Matlab and released as open-source code (https://www.mathworks.com/matlabcentral/fileexchange/67011-atom-search-optimization-aso-algorithm). This implementation has been widely utilized for various standard benchmark functions, contributing to its growing popularity. Furthermore, a Python implementation of the ASO algorithm is also accessible on GitHub (https://github.com/vlatkamihic/Atom-search-optimization).

On the other hand, several researchers have applied ASO variants to address diverse optimization problems and have generously made their code publicly available. Too & Abdullah (2020a) developed a binary version of the ASO algorithm and shared the source code online (https://github.com/JingweiToo/Binary-Atom-Search-Optimization-for-Feature-Selection). Similarly, Irudayaraj et al. (2022) proposed a chaotic-based ASO variant designed to optimize the parameters of a fractional-order PID controller in a multi-area, multi-source hybrid power system, with their source code also accessible online (https://github.com/mpremme/LFC-of-Power-System-using-Chaotic-Atom-Search-Optimization). Additionally, Python implementations of hybrid ASO-based PSO (https://github.com/ursandrew/Andrew-Xavier-Raj) and hybrid ASO-based SA (https://github.com/OmarNouih/Metaheuristic-Algorithms) are available on GitHub.

Critical analysis and discussion

The ASO algorithm is a recent advancement in the optimization field, demonstrating effectiveness and robustness in addressing a wide range of problems across various research domains. Its simplicity, flexibility, and strong memory retention enhance its performance and applicability. As shown in Algorithm 1, ASO is straightforward to implement and adaptable to various real-world optimization challenges.

As illustrated in Fig. 11, a significant portion of ASO applications is focused on engineering and design, which accounts for 62% of its usage. Environmental modeling follows, representing 11% of applications. ASO also contributes notably to emerging fields such as healthcare and medical science, as well as the Internet of Things and communication, each comprising 9% of its applications. Additionally, ASO is applied in clustering and data mining (4%) and security (4%). This distribution highlights ASO’s adaptability and applicability across diverse domains.

Figure 11 Proportional distribution of ASO applications across various areas.

Various versions of ASO have been developed to address real-world optimization problems with unique complexities and large search spaces that the original ASO or traditional methods could not effectively solve. Modifications, hybridizations, and multi-objective implementations have been introduced to navigate these challenges more efficiently, resulting in improved optimization outcomes. As shown in Fig. 12, the original ASO retains a significant presence, accounting for 49% of applications, reflecting its continued relevance. Modified ASO ranks second with 29% of applications. Hybridized ASO, which integrates ASO with other optimization techniques, is used in 20% of cases, showcasing its potential to enhance optimization performance. Finally, multi-objective ASO is applied in 1% of cases.

Figure 12 Proportional distribution of ASO and its variants.

The integration of ASO into HRES optimization has demonstrated significant benefits in enhancing energy efficiency and reliability (e.g., Rao, Srikanth Goud & Rami Reddy, 2021; Khokhar, Dahiya & Parmar, 2021; Amarendra, Srinivas & Rao, 2021; Bagade & Rampure, 2022). HRES, which combines renewable sources such as solar, wind, and biomass, faces complex challenges related to power generation, storage, and grid integration. ASO addresses these challenges by optimizing critical aspects of HRES design and operation. Additionally, ASO’s application to proportional-integral-derivative (PID) controllers has significantly advanced control system optimization (e.g., Ranjitha, Ponnurangam & Rajapandiyan, 2021; Dey, Banerjee & Dey, 2022; Irudayaraj et al., 2020; Khokhar, Dahiya & Parmar, 2020; Chandan Kumar Shiva, Kumar & Kumar, 2020). By efficiently exploring and exploiting the parameter space, ASO excels in fine-tuning PID controllers, which is essential for maintaining system stability and performance across diverse applications.

“Recent Variants of Atom Search Optimization” of the review on ASO variants highlights the significant role of various techniques in enhancing ASO. These include opposition-based learning, Lévy flight, chaotic maps, Laplacian operator-based methods, fractional calculus-based approaches, adaptive strategies, quantum-inspired techniques, mutation strategies, crossover operators, and multi-strategy approaches. Furthermore, ASO can be effectively combined with other metaheuristic algorithms from physics-based and swarm intelligence fields. Modified and hybrid approaches address ASO’s challenges, such as vulnerability to local optima, reduced population diversity, and an imbalance between exploration and exploitation. These advanced variants often deliver superior performance through the precise application of these innovative techniques.

Several ASO variants have been explored for feature selection due to their ability to address diverse challenges. As shown in Table 4, binary ASO is particularly effective for discrete feature selection by employing binary encoding for decision variables, enabling efficient selection from large feature sets. Chaotic ASO enhances exploration and mitigates local optima by introducing chaotic dynamics, making it well-suited for high-dimensional problems. Hybrid ASO integrates ASO with other techniques to improve robustness and adaptability, especially for noisy datasets and multi-objective problems. These variants highlight ASO’s flexibility and underscore the importance of considering the problem’s nature, dimensionality, optimization type, and the balance between exploration and exploitation when selecting an algorithm for feature selection.

Table 4 ASO variants for feature selection.

Authors	ASO variant	Year	
Too & Abdullah (2020b)	Chaotic ASO	2020	
Too & Abdullah (2020a)	Binary ASO	2020	
Borhade & Nagmode (2020)	Hybrid ASO	2020	
Wu, Ren & Hao (2021)	Binary ASO	2021	
Halhalli, Sugave & Jagdale (2021)	Hybrid ASO	2021	
Menaga & Revathi (2021)	Fractional-ASO	2021	
Sun et al. (2021b)	Multi-strategy ASO	2021	
Shyjith, Maheswaran & Reshma (2021)	Hybrid ASO	2021	
Hammadi & Qasim (2022)	Binary ASO	2022	
Dabbu et al. (2022)	Hybrid ASO	2022	
Ghimire et al. (2022)	Standard ASO	2022	
Duan et al. (2022b)	Standard ASO	2022	
Mahapatra, Singh & Kumar (2022)	Quantum ASO	2022	
Alhassan & Zainon (2022)	Hybrid ASO	2022	
Deore & Bhosale (2023)	Hybrid ASO	2023	
Navin et al. (2023)	Hybrid ASO	2023	
Musthafa et al. (2023)	Chaotic ASO	2023	
Lu et al. (2023)	Hybrid ASO	2023	
Borhade et al. (2023)	Hybrid ASO	2023	
Chattopadhyay et al. (2023)	Hybrid ASO	2023	
Anitha & Aravindhar (2024)	Levy flight-based ASO	2024	
Singh & Kumar (2024)	Hybrid ASO	2024	
Maazalahi & Hosseini (2024)	Hybrid ASO	2024	
Pitchandi, Nivaashini & Grace (2024)	Hybrid ASO	2024	
Ali et al. (2024)	Standard ASO	2024	

As mentioned earlier, the ASO algorithm has gained widespread recognition for solving various problems since its introduction. Its effectiveness can be attributed to its simple inspiration, few control parameters, and flexible search behavior. However, despite being a stochastic optimization technique, it has some limitations and drawbacks that impact its efficiency.

The main drawback of the ASO algorithm is based on the NFL theorem (Wolpert & Macready, 1997), a common challenge for many optimization methods. According to the NFL theorem, no optimization algorithm can consistently outperform others across all types of problems or even different instances of the same problem. As a result, the convergence of the ASO algorithm heavily depends on the specific features of the problem’s search space. Therefore, when dealing with an unknown search space, it is crucial to adapt the ASO algorithm or combine it with other methods to solve the optimization problem effectively.

The second disadvantage is related to the update mechanism in ASO. The velocity update equation in ASO depends solely on individual atom velocities and accelerations, which can lead to reduced population diversity and limited information exchange, as it overlooks the influence of atom positions. Additionally, the attractive force in ASO may cause atoms to cluster early on, leading to premature convergence. Conversely, the repulsive force may not be strong enough to help atoms escape local optima later in the process. This premature convergence impedes exploration, while the inability to escape local optima limits exploitation, thus restricting ASO’s effectiveness in solving complex optimization problems.

The third limitation concerns the problem domain. As noted earlier, the original ASO was designed for optimization problems with continuous search spaces, single objectives, and unconstrained objective functions. However, to address a broader range of problems, including binary, combinatorial, dynamic, many-objective, and discrete challenges, ASO requires an expanded scope of applicability.

Another disadvantage of ASO’s performance is its sensitivity to dimensionality. As the number of variables in a problem increases, the algorithm’s effectiveness tends to decrease, often resulting in slower convergence and higher computational complexity. This issue arises because higher-dimensional spaces present more challenges for the search process, making it harder for the algorithm to explore and exploit the search space effectively.

Lastly, one of the significant limitations of ASO is its lack of diversity control, which leads to premature convergence. During the early stages, ASO operators select only the best solution per generation, neglecting other solutions in the population. This reduces diversity, causing the algorithm to stagnate at local minima and preventing exploration of the entire search space. As a result, the algorithm converges toward the optimal solution too quickly, overlooking that reaching the global optimum often requires exploring suboptimal solutions.

Conclusion and further works

This article provides a comprehensive review of the Atom Search Optimization (ASO) algorithm and its diverse applications across various research domains. The study encompasses all relevant research contributions accessible to the authors, focusing on ASO and its variants, identified through an analysis of approximately 150 journal and conference articles indexed in Web of Science and Scopus at the time of writing. Despite its relatively recent introduction to the field of metaheuristics, this review demonstrates that ASO has rapidly expanded in application, both in its standard form and through modified and hybrid implementations with other algorithms. The notable growth of ASO is reflected in the exponential increase in published research, with new articles appearing regularly. Moreover, the reviewed articles reveal that the ASO algorithm has primarily been applied to solving global optimization problems, leaving its potential for addressing other challenges as an open area for future exploration. By leveraging the strengths of the ASO algorithm, complex optimization problems in these fields could be solved effectively and efficiently.

This review article aims to provide scholars with an invaluable resource, particularly those working or planning to work in the field of metaheuristic algorithms in general and the ASO algorithm. By providing insights into how the ASO algorithm can be applied to address challenges across various domains, this article seeks to contribute to advancing research in this area. Furthermore, the reviewed studies identify several promising research avenues that warrant further exploration, including: The success of ASO in addressing a diverse array of problems will likely motivate researchers to apply the algorithm to additional fields, including smart homes, robotics, work scheduling, chemical engineering, signal denoising, economic forecasting, the DNA fragment assembly problem, knowledge discovery, smart grids, and energy storage devices.

Improving the ASO algorithm is crucial for efficiently addressing complex and evolving optimization problems. Modifications should focus on adapting ASO to meet the diverse needs of interdisciplinary applications. Studies suggest dividing the population into sub-groups with different ASO strategies to maintain diversity (Zhang, Wen & Wang, 2022), and parallel ASO implementations can accelerate convergence without compromising solution quality (Liang et al., 2022).

Combining ASO with other metaheuristic algorithms presents a significant opportunity to advance the field. By integrating ASO with recently published algorithms such as GO (Zhang et al., 2023b), PKO (Bouaouda et al., 2024), PSA (Gao, 2023), and others, hybrid models can be developed that leverage the unique strengths of each method. This approach aims to deliver more effective solutions than individual techniques alone.

Adapting ASO to handle high-dimensional and large-scale optimization problems effectively is another important research area. To address this, dimensionality reduction methods such as principal component analysis (PCA) (Alomari et al., 2022) can be applied to specific problems to improve efficiency. Additionally, other scalability and transformation techniques should be explored.

Parameter tuning is a critical and often challenging aspect of using metaheuristic algorithms. In many metaheuristic methods, the initial parameter values are kept constant during the algorithm’s execution. However, adopting an adaptive parameter tuning strategy presents a promising direction for further exploration. Developing intelligent and adaptive techniques for parameter tuning in ASO remains an open research area.

ASO is barely applied to multi-objective problems in scientific and engineering fields, with most studies focusing on single-objective optimization. Future research should focus on developing multi-objective variants of ASO to expand its applicability and address unexplored gaps while leveraging ASO’s strengths in single-objective optimization for broader practical use.

Abbreviations

AIC Automatic Image Captioning

ANN Artificial Neural Network

BASO Binary Atom Search Optimization

BP Back Propagation

BWO Black Widow Optimization

CASO Chaotic Atom Search Optimization

CCD Central Composite Design

CNN Convolutional Neural Network

DA Dragonfly Algorithm

DG Distributed Generation

DNN Deep Neural Network

EBO Ecogeography-Based Optimization

ES Evolutionary Strategy

FEM Finite Element Method

FASO Fast Atom Search Optimization

FOPID Fractional Order Proportional Integral Derivative

FPA Flower Pollination Algorithm

GMPP Global Maximum Power Point

GSA Gravitation Search Algorithm

GWO Grey Wolf Optimizer

HHO Harris Hawk Optimization

HMG Hybrid Micro-Grid

HOS-LSDA Higher Order Statistics-Latent Space Data Analysis

HPS Hybrid Power Systems

HRES Hybrid Renewable Energy Sources

IASO Improved Atom Search Optimization

JA-FPA Jaya Flower Pollination Algorithm

LFC Load Frequency Control

MFO Moth-Flame Optimizer

M-PID Multistage Proportional-Integral-Derivative

MMG Multi-Microgrid

MO-QASO Multi-Objective Quantum Atom Search Optimization

MCCHP Micro Combined Cooling, Heating, and Power Generation

NPC Net Present Cost

OPF Optimal Power Flow

PEM Polymer Exchange Membrane

PEVs Plug-in Electric Vehicles

PIDA Proportional-Integral-Derivative-Acceleration

PID Proportional-Integral-Derivative

PLS Power Loss Sensitivity

QSA Queuing Search Algorithm

RDS Radial Distribution system

RE Renewable Energy

SHE Selective Harmonic Elimination

SSA Salp Swarm Algorithm

STATCOM Static Synchronous Compensator

SVM Support Vector Machine

THD Total Harmonic Distortion

TID Tilt Integral Derivative

UPQC Unified Power Quality Conditioner

WOA Whale Optimization Algorithm

Additional Information and Declarations

Competing Interests

The authors declare that they have no competing interests.

Author Contributions

Mohammed A. El-Shorbagy conceived and designed the experiments, performed the experiments, analyzed the data, performed the computation work, prepared figures and/or tables, authored or reviewed drafts of the article, and approved the final draft.

Anas Bouaouda conceived and designed the experiments, performed the experiments, analyzed the data, performed the computation work, prepared figures and/or tables, authored or reviewed drafts of the article, and approved the final draft.

Laith Abualigah performed the experiments, analyzed the data, prepared figures and/or tables, authored or reviewed drafts of the article, and approved the final draft.

Fatma A. Hashim performed the experiments, analyzed the data, prepared figures and/or tables, authored or reviewed drafts of the article, and approved the final draft.

Data Availability

The following information was supplied regarding data availability:

This is a literature review.

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
