# Peer review of "Atom Search Optimization: a comprehensive review of its variants, applications, and future directions"

_PeerJ Computer Science, doi:10.7717/peerj-cs.2722_

## Round 0.1 · original submission · Minor Revisions

Dear Authors,

Your paper has been reviewed. Based on the reviewers’ reports, minor revisions are needed before it is considered for publication in PEERJ Computer Science. In particular:

1) You should include existing studies in your revised manuscript to emphasize the factors in choosing nature-inspired optimization algorithms to find (feature selection) the feature set that has to be used to solve specific problems.

2) While the article highlights the success of ASO in general, you have to include comparative results that numerically support this success by adding a simple table or graph comparing ASO to popular metaheuristic algorithms.

3) Although the paper mentions future research directions, these recommendations remain at a general level. You must provide more specific directions, such as which areas ASO needs further development.

Reviewer 1 ·

Basic reporting

No Comment

Experimental design

Provide a clear explanation of your taxonomy. Consider including visual representations, such as flowcharts or diagrams, to illustrate how different ASO variants and applications fit within the taxonomy.

Validity of the findings

Elaborate on the common challenges researchers face when selecting algorithms for real-world problems.

Reviewer 3 ·

Basic reporting

In this paper, the authors provide a comprehensive review of the Atom Search Optimization (ASO) algorithm, a metaheuristic optimization technique introduced in 2019 that is based on the principles of molecular dynamics. ASO models the natural behavior of atoms interacting through forces derived from the Lennard-Jones potential and bond-length potentials. Since its inception, ASO has been successfully applied to a wide range of continuous optimization problems across various fields. Despite its growing use and the expanding literature, a thorough evaluation of its implementations has been lacking. This paper addresses this gap by reviewing half a decade of advancements in ASO research and discussing key variants, foundational principles, and notable achievements. It covers both single- and multi-objective optimization problems and presents a taxonomy to guide further research in ASO. The paper also highlights challenges faced by researchers in selecting suitable algorithms for real-world problems, offering valuable insights into the design of ASO variants and other metaheuristic algorithms. The quality of some Figures should be significantly improved. And illustrated in the discussion section with more details.

The literature review is inadequate and fails to cover key relevant works. A more thorough and up-to-date review of the related literature is essential and should be discussed.

“ Mandour, S., Gamal, A. and Sleem, A. (2024) “Mantis Search Algorithm Integrated with Opposition-Based Learning and Simulated Annealing for Feature Selection”, Sustainable Machine Intelligence Journal, 8, pp. (5):56–98. doi:10.61356/SMIJ.2024.8300. ; Salem, S. (2023) “An Improved Binary Quadratic Interpolation Optimization for 0-1 Knapsack Problems”, Sustainable Machine Intelligence Journal, 4, pp. (1):1–11. doi:10.61185/SMIJ.2023.44101.; Saber, S. and Salem, S. (2023) “High-Performance Technique for Estimating the Unknown Parameters of Photovoltaic Cells and Modules Based on Improved Spider Wasp Optimizer”, Sustainable Machine Intelligence Journal, 5, pp. (2):1–14. doi:10.61185/SMIJ.2023.55102.”

The research methods' limitations should be further discussed in the paper. To improve the readability of symbols, I recommend the authors make a table at the end of the introduction to summarize the meaning of all symbols in their study. The experimental and analysis part of this work needs to be further expanded and improved. discussion should be rewritten to summarize the findings/significance of the work.
The methodology section lacks detail and rigor. The authors should provide a more comprehensive description of their methods, including justification for chosen techniques and parameters. The conclusion section should be expanded. Discuss the future concerning the research state of progress and its limitations.

Experimental design

In this paper, the authors provide a comprehensive review of the Atom Search Optimization (ASO) algorithm, a metaheuristic optimization technique introduced in 2019 that is based on the principles of molecular dynamics. ASO models the natural behavior of atoms interacting through forces derived from the Lennard-Jones potential and bond-length potentials. Since its inception, ASO has been successfully applied to a wide range of continuous optimization problems across various fields. Despite its growing use and the expanding literature, a thorough evaluation of its implementations has been lacking. This paper addresses this gap by reviewing half a decade of advancements in ASO research and discussing key variants, foundational principles, and notable achievements. It covers both single- and multi-objective optimization problems and presents a taxonomy to guide further research in ASO. The paper also highlights challenges faced by researchers in selecting suitable algorithms for real-world problems, offering valuable insights into the design of ASO variants and other metaheuristic algorithms.

Validity of the findings

The methodology section lacks detail and rigor. The authors should provide a more comprehensive description of their methods, including justification for chosen techniques and parameters. The conclusion section should be expanded. Discuss the future concerning the research state of progress and its limitations.

Additional comments

No more comments.

Reviewer 4 ·

Basic reporting

The above paper solves the " Atom Search Optimization: A comprehensive review of its variants, applications, and future directions ".

The paper is well-written. This topic is very good and timely, and the proposed algorithm is skillfully developed, but the paper has some problems, authors should revise them.

After considering the following minor revisions, the paper will be accepted.

1. Add the contents in the abstract of the chapter.

2. The paper should check the professional English proofreading check to fix the language issues.

3. The literature review is poor in this paper. you must review all significant similar works that have been done.

4. Modify the proposed pseudo-code of the ASO algorithm (Algorithm 1). Learn how to write pseudo-code in different papers. Please read it.

5. The conclusion of the paper should be corrected. Conclusion should include problem, contribution and results concisely.

Experimental design

As above

Validity of the findings

As above

Additional comments

As above

Annotated reviews are not available for download in order to protect the identity of reviewers who chose to remain anonymous.

Reviewer 5 ·

Basic reporting

No comment

Experimental design

No comment

Validity of the findings

No comment

Additional comments

The paper presents a comprehensive review of the Atom Search Optimization (ASO) algorithm, covering its development, applications, and future research directions. The basic principles of ASO are explained in detail in the paper. The classification of different variants in the literature allows readers to gain a deep understanding of the algorithm and guides them to related studies. I believe that this systematic approach will be an exemplary study for literature reviews. I think the article is generally well-organized, written, and understandable to readers. It is extremely valuable that the paper includes applications of the ASO algorithm in both single and multi-objective optimization problems and highlights its successes in different areas. The paper critically examines the current implementations of the ASO algorithm and provides suggestions for further development areas, increasing the academic value of the study.
- It is recommended that existing studies be included to emphasize the factors in choosing nature-inspired optimization algorithms to find (feature selection) the feature set that will be used to solve certain problems.
- While the article highlights the success of ASO in general, is it possible to include comparative results that numerically support this success? Adding a simple table or graph comparing ASO to popular metaheuristic algorithms could make it easier for readers to evaluate the algorithm.
-Although the paper mentions future research directions, these recommendations remain at a general level. It would be useful to provide more specific directions, such as which areas ASO needs further development.
As a result, this paper is a very valuable study that will strengthen the place of the ASO algorithm in the literature, inspire researchers, and provide a basis for potential applications in different disciplines.

---

## Round 0.2 · accepted · Accept

Dear Authors,

Your paper has been revised. It has been accepted for publication in PEERJ Computer Science. Thank you for your fine contribution.

Reviewer 1 ·

Basic reporting

no comment

Experimental design

no comment

Validity of the findings

no comment

Reviewer 5 ·

Basic reporting

It has been observed that the authors have implemented the suggested corrections.

Experimental design

-

Validity of the findings

-

Additional comments

There is no problem in accepting the article for publication in the journal in this form.